# Intestinal iron absorption is appropriately modulated to match physiological demand for iron in wild-type and iron-loaded *Hamp* (hepcidin) knockout rats during acute colitis

Shireen R. L. Flores[1], Savannah Nelson[1], Regina R. Woloshun[1], Xiaoyu Wang[1¤a], Jung-Heun Ha[1¤b¤c], Jennifer K. Lee[1], Yang Yu[1], Didier Merlin[2,3], James F. Collins[1]*

**1** Food Science & Human Nutrition Department, University of Florida, Gainesville, FL, United States of America, **2** Center for Diagnostics and Therapeutics, Institute for Biomedical Science, Georgia State University, Atlanta, GA, United States of America, **3** Atlanta Veterans Affairs Medical Center, Decatur, GA, United States of America

¤a Current address: Key Laboratory of Precision Nutrition and Food Quality, College of Food Science and Nutritional Engineering, China Agricultural University, Beijing, China
¤b Current address: Department of Food Science and Nutrition, Dankook University, Cheonan, Korea
¤c Current address: Research Center for Industrialization of Natural Nutraceuticals, Dankook University, Cheonan, Korea
* jfcollins@ufl.edu

**Data Availability Statement:** All relevant data are within the manuscript and its Supporting Information files.

## Abstract

Mucosal damage, barrier breach, inflammation, and iron-deficiency anemia (IDA) typify ulcerative colitis (UC) in humans. The anemia in UC appears to mainly relate to systemic inflammation. The pathogenesis of this 'anemia of inflammation' (AI) involves cytokine-mediated transactivation of hepatic *Hamp* (encoding the iron-regulatory hormone, hepcidin). In AI, high hepcidin represses iron absorption (and iron release from stores), thus lowering serum iron, and restricting iron for erythropoiesis (causing anemia). In less-severe disease states, inflammation may be limited to the intestine, but whether this perturbs iron homeostasis is uncertain. We hypothesized that localized gut inflammation will increase overall iron demand (to support the immune response and tissue repair), and that hepatic *Hamp* expression will decrease in response, thus derepressing (i.e., enhancing) iron absorption. Accordingly, we developed a rat model of mild, acute colitis, and studied iron absorption and homeostasis. Rats exposed (orally) to DSS (4%) for 7 days had intestinal (but not systemic) inflammation, and biomarker analyses demonstrated that iron utilization was elevated. Iron absorption was enhanced (by 2-3-fold) in DSS-treated, WT rats of both sexes, but unexpectedly, hepatic *Hamp* expression was not suppressed. Therefore, to gain a better understanding of regulation of iron absorption during acute colitis, *Hamp* KO rats were used for further experimentation. The severity of DSS-colitis was similar in *Hamp* KOs as in WT controls. In the KOs, increased iron requirements associated with the physiological response to colitis were satisfied by mobilizing hepatic storage iron, rather than by increasing absorption of enteral iron (as occurred in WT rats). In conclusion then, in both sexes and genotypes of rats, iron absorption was appropriately modulated to match physiological demand for dietary iron during acute intestinal inflammation, but regulatory mechanisms may not involve hepcidin.

**Funding:** This investigation was funded by grants R01 DK074867 from the National Institute of Diabetes and Digestive and Kidney Diseases (NIDDK) (JFC), and R01 DK109717 from NIDDK and the Office of Dietary Supplements (JFC). The funders had no role in study design, data collection and analysis, decision to publish, or preparation of the manuscript.

## Introduction

Regulating absorption of dietary iron to match needs is critical, since humans (and most other mammals) do not possess specific, regulated iron-excretory systems. Careful experimentation over the past several decades has described (patho)physiological conditions that are associated with modulation of intestinal iron absorption [1–5]. For example, iron absorption is enhanced during iron depletion and hypoxia, and when erythropoiesis is stimulated. Conversely, when iron stores are replete and during inflammation, intestinal iron transport is blunted. It is now clear that the peptide hormone, hepcidin, is the principal regulator of intestinal iron absorption. Although the *Hamp* gene (encoding hepcidin) is transcriptionally active in multiple tissues (https://www.genecards.org/cgi-bin/carddisp.pl?gene=HAMP#expression), blood-borne hepcidin is thought to be exclusively derived from the liver [6]. Circulating hepcidin regulates serum iron flux by modulating cell-surface expression of the iron exporter ferroportin 1 (FPN1) in duodenal enterocytes (which absorb enteral iron), and macrophages of the reticulo-endothelial [RE] system and hepatocytes (which store iron) [7]. Hepcidin fulfills the functions of the so called 'stores' and 'inflammatory' regulators of iron absorption, as *Hamp* expression in hepatocytes is induced when body iron levels are elevated and during systemic inflammation [8]. In iron overload, increased hepcidin decreases FPN1 protein levels, blunting iron absorption and decreasing serum iron, which reduces body iron burden. Hepcidin-mediated iron sequestration also potentiates the acute-phase response to infection and inflammation. This peptide hormone also fulfills the functions of the 'iron deficiency / hypoxia', and 'erythroid' regulators of iron absorption, as *Hamp* expression is strongly downregulated during iron depletion (with possible concurrent hypoxia), and when erythroid demand is elevated. Lower circulating hepcidin levels lead to increases in FPN1 protein levels, enhancing iron absorption and increasing serum iron, thus reestablishing normal body iron levels, and providing extra iron to support hemoglobin synthesis in developing erythrocytes. In sum, hepcidin is considered the main regulator of systemic iron homeostasis, functioning via its ability to regulate serum iron concentrations by modulating dietary iron absorption and iron release from stores.

Disruption of iron homeostasis commonly occurs in inflammatory bowel diseases (IBD), including, most commonly, Crohn's disease and ulcerative colitis (UC) [9–11]. Dysregulation of *Hamp* expression may contribute to the pathogenesis of these disorders [12, 13]. Active IBD is likely to cause breach of the intestinal barrier and release of antigenic molecules (e.g., bacterially derived lipopolysaccharide) into the blood stream, which may result in systemic inflammation. Pro-inflammatory cytokines (e.g., IL-6) transactive hepatic *Hamp* leading to increased levels of circulating hepcidin. High hepcidin blunts iron absorption and decreases serum iron, possibly leading to iron-restricted erythropoiesis and consequent anemia, a condition referred to as the 'anemia of inflammation' (AI) [14]. AI is a multifaceted comorbidity that complicates treatment of IBD [15]. Although molecular perturbations underlying AI have been described [16], whether dysregulation of iron homeostasis occurs in less severe disease states where barrier function remains intact and systemic inflammation does not occur is unclear. This latter situation is of clinical significance, representing times between active disease flares in IBD patients and individuals with milder forms of GI disease. The current investigation was thus designed to test the hypothesis that localized intestinal inflammation also alters iron absorption. Specifically, we predicted that acute intestinal inflammation would increase iron demand (to support repair of damaged tissue and immune cell proliferation), which will cause repression of *Hamp*, thus decreasing blood hepcidin levels and enhancing iron absorption (and iron release from stores). Accordingly, a rat model of acute colitis was developed, and iron homeostasis was assessed. Notable experimental outcomes provided the rationale for pursuing

experiments designed to further understand how (if at all) hepcidin influences iron absorption during acute colitis. This was accomplished by performing parallel experiments in global *Hamp* KO rats. Outcomes showed that acute, localized intestinal inflammation had distinct effects on iron absorption (and iron homeostasis), depending upon the sex and genotype/phenotype of experimental rats used in the study.

## Materials and methods

### Animal procedures

All animal procedures were approved by the University of Florida IACUC. Irradiated Teklad Rodent Diet #2919 (200 mg/kg iron) (Envigo; Hayward, CA) was provided *ad libitum* to all rats throughout all experiments, as was purified water. Food and water intake were estimated daily in experimental rats by weighing the amount of food necessary to refill food hoppers and measuring the volume of water required to refill water bottles. The age of rats used for various experiments is provided in the figures or figure legends, (since it varied [slightly] across different experiments). A detailed description of the generation of *Hamp* KO Sprague-Dawley rat lines and histological analysis of liver, spleen and other tissues from WT (control) and KO rats are provided below.

### Generation of the *Hamp* knockout (KO) rat

*Hamp* KO rats were generated by Transposagen Biopharmaceuticals (Lexington, KY). Briefly, a TALEN pair with the recognition sequence 5'– TCTGCTTTCACAGACGAGacagactacg gctctgcaGCCTTGGCATGGGGCAGA–3' targeting the second exon of the rat *Hamp* gene was generated using standard methods [17] and cloned into proprietary expression vectors. TALEN mRNA encompassing the forward and reverse half sites was generated using the mMessage mMachine *in vitro* transcription kit (Cat# AM1345; ThermoFisher). TALEN mRNA was injected into pronuclei of Sprague Dawley rat embryos as detailed in Tesson, et al. [18], and standard protocols were utilized to subsequently produce live offspring. Genomic DNA from potential founder rats was screened by PCR using primers that spanned the targeted region. PCR products were validated by sub-cloning and sequencing.

After transfer from Transposagen Biopharmaceuticals, a breeding colony was established on the University of Florida campus. *Hamp*$^{-/-}$ rats were interbred with rats carrying the same deletion. Similarly, WT rats were also interbred. Rats were genotyped just prior to weaning by extracting genomic DNA from tail snip samples (Quick-DNA Miniprep Plus Kit; Zymogen) and performing genomic PCR analysis (primer sequences are listed in Table 1).

### Histological analysis of liver, spleen and other tissues

Tissues were fixed in 10% neutral buffered formalin for at least 20 hours and then subjected to routine histological processing, including H&E staining of sections. For analysis of the spleen morphology, the proportion of red versus white pulp in tissue cross sections was assessed (by a

**Table 1. Sequences of primers used for genotypic analysis of the four *Hamp* KO rat lines.**

| Deletion Sizes | Mutations | Primer Sequences (5'-3') |
|---|---|---|
| Less than 50 bp | Del 15, Del 20 Ins 3 | F–CTTAGGAAGGGACCCGTGTT |
| | | R–GACAAGGCTCTTGGCTCTCT |
| Greater than 100 bp | Del 169, Del 230 Ins 1 | F–CCACTTCCCCATCTGCATTT |
| | | R–GCAGCACATCCCACACTTTG |

blinded observer) by quantifying the area of periarterial lymphatic sheaths (PALS), representing white pulp, and comparing it to total cross-sectional area. Additionally, some tissues sections were deparaffinized and stained for hemosiderin ferric iron deposits using Perls' Prussian blue stain with or without cobalt enhancement (as necessary).

## Induction and assessment of colitis

One goal of this investigation was to assess the impact of GI tract-restricted inflammation on iron absorption, without the confounding influence of intestinal barrier breach and concomitant systemic inflammation (which is accompanied by pro-inflammatory cytokine-mediated transactivation of *Hamp*). We thus selected the dextran sodium sulfate (DSS) method to induce epithelial damage and intestinal inflammation. The DSS model of colitis faithfully recapitulates some aspects of human IBD [19]. Moreover, this experimental approach was considered advantageous for our studies since it is rapid, simple, reproducible, and most importantly, it allows precise control of the extent of mucosal damage and concomitant localized or systemic inflammation. We chose rats for this investigation given our longstanding belief that rats may better model certain aspects of iron homeostasis in humans (than do, for example, mice). Since previous studies demonstrated that environmental, genetic and species-specific factors influence the pathophysiological response to DSS exposure, pilot studies were performed in which adult Sprague-Dawley rats were exposed to 3, 3.5 or 4% Dextran Sulfate Sodium (DSS) (Colitis Grade [36,000–50,000 Da]; CAS # 9011-18-1; MP Biomedicals; Irvine, CA) in drinking water for 7 days. Ultimately, 4% DSS was chosen for subsequent experiments since a mild, acute colitis was noted with this concentration of DSS. All rats used for the experiments described herein were 10–12 weeks of age and were terminated after 7 days of DSS treatment. Clinical signs of colitis, including weight loss, diarrhea, and blood in the stool (hemoccult), were quantified daily for the 7-day treatment period. These data were used to calculate an overall Disease Activity Index score (DAI), using a previously described scoring system [20]. Furthermore, histological analyses were performed on H&E stained, transverse sections from distal colon, as previously described [20, 21]. The extent of inflammatory infiltrate (neutrophils & lymphocytes) and shortening of the crypts were assessed in 4 randomly selected areas of each sample by a blinded (experienced) observer, and values from 0 (least severe) to 4 (most severe) were assigned. Individual area scores for each animal were averaged and then combined to calculate an average histological score for each experimental group.

## Measurement of blood iron-related parameters

After exposure to $CO_2$ gas in a sealed chamber (causing narcosis), blood was collected by cardiac puncture, and rats were subsequently killed by thoracotomy followed by cervical dislocation. Hemoglobin (Hb) was measured with a hemoglobin analyzer (HemoCue). To quantify hematocrit (Hct), blood was pipetted into capillary tubes (ThermoFisher Inc.; Waltham, MA) and spun for 2 mins using a Readacrit centrifuge (Clay Adams; Franklin Lakes, NJ). Aliquots of whole blood were sent to the University of Florida, College of Veterinary Medicine Diagnostic Laboratory for complete blood count (CBC) analysis. Plasma was obtained by centrifugation of heparinized blood at 2,000 x *g* at 4° C for 10 min. Plasma nonheme iron concentration and total iron-binding capacity (TIBC) were determined as described previously [22, 23]. Transferrin saturation (TSAT) was calculated as ([plasma iron/TIBC] x 100).

## Nonheme iron determination

Tissue nonheme iron concentrations were measured by a standard, colorimetric procedure, as described previously [24]. Briefly, a sample of snap-frozen tissue (~50 mg) was digested in an

acid solution (3 M hydrochloric acid plus 10% [w/v] trichloroacetic acid) and incubated for 20 h at 65°C. An aliquot of the digested supernatant was then mixed with a chromogen reagent [0.1% (w/v) bathophenanthroline sulfonate plus 1% (v/v) thioglycolic acid], milli Q water and sodium acetate at a ratio of 1:5:5. The absorbance (OD) was then read at 535 nm on a Synergy H1 plate reader (BioTek) and compared to a certified iron reference solution (cat. # SI124-100; ThermoFisher).

## Quantitative RT-PCR analysis

Expression of hepcidin mRNA in the liver was assessed by qRT-PCR, using a standard protocol based upon the $\Delta\Delta_{CT}$ approach, as detailed previously [25, 26]. Primer sequences were as follows: hepcidin forward (5′-GGCAGAAAGCAAGACTGATGAC- 3′) and reverse (5′-ACA GGAATAAATAATGGGGCG-3′); cyclophilin forward (5′-CTTGCTGCAATGGTCAACC-3′) and reverse (5′-TGCTGTCTTTGGAACTTTGTCTGC- 3′). Hepcidin expression was normalized to cyclophilin expression (which varied little between experimental samples).

## Measurement of serum IL-6 and other inflammatory biomarkers

During systemic inflammation, increased levels of circulating pro-inflammatory cytokines, including most potently IL-6, transactivate *Hamp* in hepatocytes contributing to the development of AI [27]. Serum IL-6 was thus quantified via ELISA in experimental rats, following the manufacturer's instructions (ab234570; Abcam; Cambridge, MA). Briefly, an antibody mix was added to experimental samples or standards in a 96 well plate. After incubation, the wells were washed to remove unbound material. Upon further reagent addition, a visible light signal was generated in proportion to the amount of bound analyte (i.e., IL-6), and the signal intensity was measured spectrophotometrically at 450 nm. Additionally, to further consider the possibility that DSS treatment resulted in systemic inflammation, a rat protein cytokine analysis kit was utilized (Rat Cytokine Antibody Array C1 Series 2000; AAR-CYT-1-8; RayBiotech, Inc.; Norcross, GA). Briefly, rat serum was mixed with a cell lysis buffer and then centrifuged at 1,500 x *g* to remove cell debris. The supernatant was collected (for the assay) and protein concentration was determined. The membranes were first incubated in a blocking buffer, and then equal amounts of serum proteins were added to each array. After a 2-hour incubation at room temperature, the membranes were further processed and analyzed according to the manufacturer's instructions.

## Quantification of intestinal iron absorption in experimental rats

Iron absorption was quantified by a radioactive tracer method, essentially as described previously [26, 28]. To prepare rats for oral, intragastric gavage, animals were acclimated to human contact and the necessary experimental holds for at least 10 min/day starting 7 days before the DSS treatment period. Acclimatization then continued daily during the experimental period. Two days prior to the end of experiments (on day 5 of DSS treatment), rats were fasted overnight (but provided drinking water [with or without DSS]). The next day (on day 6), rats were administered 250 μL of a solution containing 10 mmol/L HCl, 200 μmol/L $FeSO_4$ and 3 μCi $^{59}FeCl_3$ (Perkin Elmer; Waltham, MA) by oral, intragastric gavage. Immediately after dosing, initial radioactivity in each animal was determined at a fixed distance using a Ram DA Counter with a PM-11 tube (Rotem Industries; Arava, Israel). Rats were then allowed unrestricted access to food and water. Twenty-four hours later (on day 7), the amount of radioactivity remaining in each rat was determined by whole-body counting (using the same technique as described above). Rats were subsequently killed, and the entire GI tract (esophagus to anus), blood, liver, spleen, heart, pancreas, bone marrow, and muscle were harvested for

individual radioactivity measurements using a WIZARD$^2$ automatic gamma counter (Perkin Elmer). The percentage of $^{59}$Fe absorbed from the test dose was calculated by dividing the total radioactivity remaining in the rat after 24 h (minus radioactivity remaining in the GI tract) by the initial radioactivity for that animal (and multiplying by 100).

## Statistical analyses

Data are presented as box plots displaying the minimum, the lower [25$^{th}$ percentile], the median [50$^{th}$ percentile], the upper [75$^{th}$ percentile] and the maximum ranked sample (unless indicated otherwise in the figure legend). Data were compared by using, where appropriate, a Student's unpaired t test or a one- or two-way ANOVA using Prism GraphPad (v6) software. When a significant main effect was identified with one-way ANOVA, or when a significant 2-way interaction was noted when using two-way ANOVA, Tukey's post hoc test was utilized to assess differences between individual groups. $P<0.05$ was considered statistically significant. Departures from normal distribution were checked using the D'Agostino & Pearson goodness-of-fit test. Datasets with unequal variances were log-transformed before statistical analyses were undertaken.

## Results

### Successful generation of the *Hamp* KO, Sprague-Dawley rat

Four different genetically engineered rat lines, each with different sized *Hamp* deletions (with or without concurrent insertions), were created using TALEN technology. Two larger deletions spanned exons 2 and 3 of the WT gene: the first harbored a 230 bp deletion, with a one bp insertion (Del 230, Ins 1) [spanning from the 12$^{th}$ bp of exon 2, with insertion of an 'A' bp, across intron 2, and including the first 95 bp of exon 3]; the second was a continuous 169 bp deletion (Del 169) [from the 8$^{th}$ base of exon 2, across intron 2, and including the first 30 bp of exon 3]. Two smaller deletions removed base pairs in exon 2: Del 20, Ins 3 [from the 10$^{th}$ bp, with a 'CCT' insertion, to bp 29) and Del 15 [from bp 6 to 20]. Base pair numbers are based upon NCBI Reference Sequence NM_053469.1. A schematic showing the relative positions of expected alterations in the hepcidin amino acid sequence is provided in Fig 1A. Sperm from heterozygous and homozygous null rats from all 4 mutant lines was cryopreserved and deposited at the Rat Resource and Research Center (RRRC) (University of Missouri; Columbia, MO, USA). Strain IDs are as follows: Del 230, Ins 1 (SD-*Hamp*$^{em2Jfcol\ +/-}$ [RGD:36174221], SD-*Hamp*$^{em2Jfcol\ -/-}$ [RGD:36174222] [RRRC# 909]); Del 169 (SD-*Hamp*$^{em1Jfcol\ +/-}$ [RGD: 36174030], SD-*Hamp*$^{em1Jfcol\ -/-}$ [RGD:36174220] [RRRC# 907]); Del 20, Ins 3 (SD-*Hamp*$^{em3Jfcol\ +/-}$[RGD:361 74223], SD-*Hamp*$^{em3Jfcol\ -/-}$ [RGD:36174224] [RRRC# 911]); Del 15 (SD-*Hamp*$^{em4Jfcol\ +/-}$ [RGD:36174225], SD-*Hamp*$^{em4Jfcol\ -/-}$ [RGD:36174226] [RRRC# 913]). PCR analysis was used to verify the corresponding *Hamp* mutations in all four strains (Fig 1B). Furthermore, the hepcidin mRNA transcript was essentially undetectable (by qRT-PCR) in any of the homozygous mutant strains, confirming successful ablation of the *Hamp* gene (Fig 1C).

### *Hamp* ablation in rats increased hepatic iron content and depleted splenic iron

To determine whether lack of hepcidin resulted in tissue iron loading, nonheme iron levels were quantified in 8-9-week-old WT, *Hamp*$^{+/-}$ and *Hamp*$^{-/-}$ rats of both sexes from all 4 mutant lines. Significant increases in liver nonheme iron levels were observed in all homozygous mutant lines when compared to WT controls ($p<0.0001$) (Fig 2A). Additionally, nonheme Fe levels in the spleen were significantly lower in all four *Hamp*$^{-/-}$ lines ($p<0.0001$) (Fig 2B). No significant differences (from WT rats) in nonheme iron content were observed in

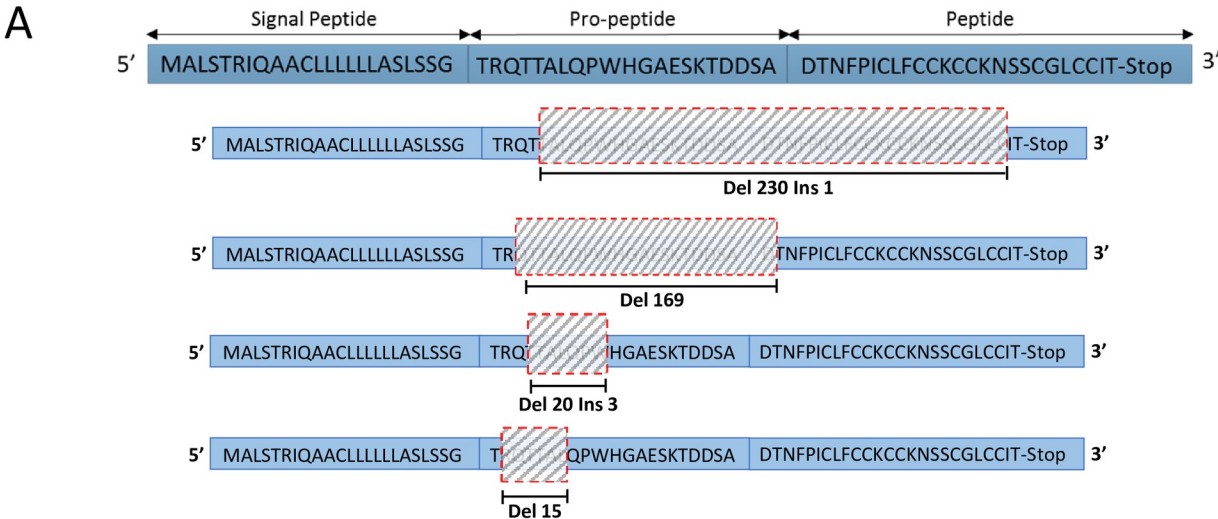

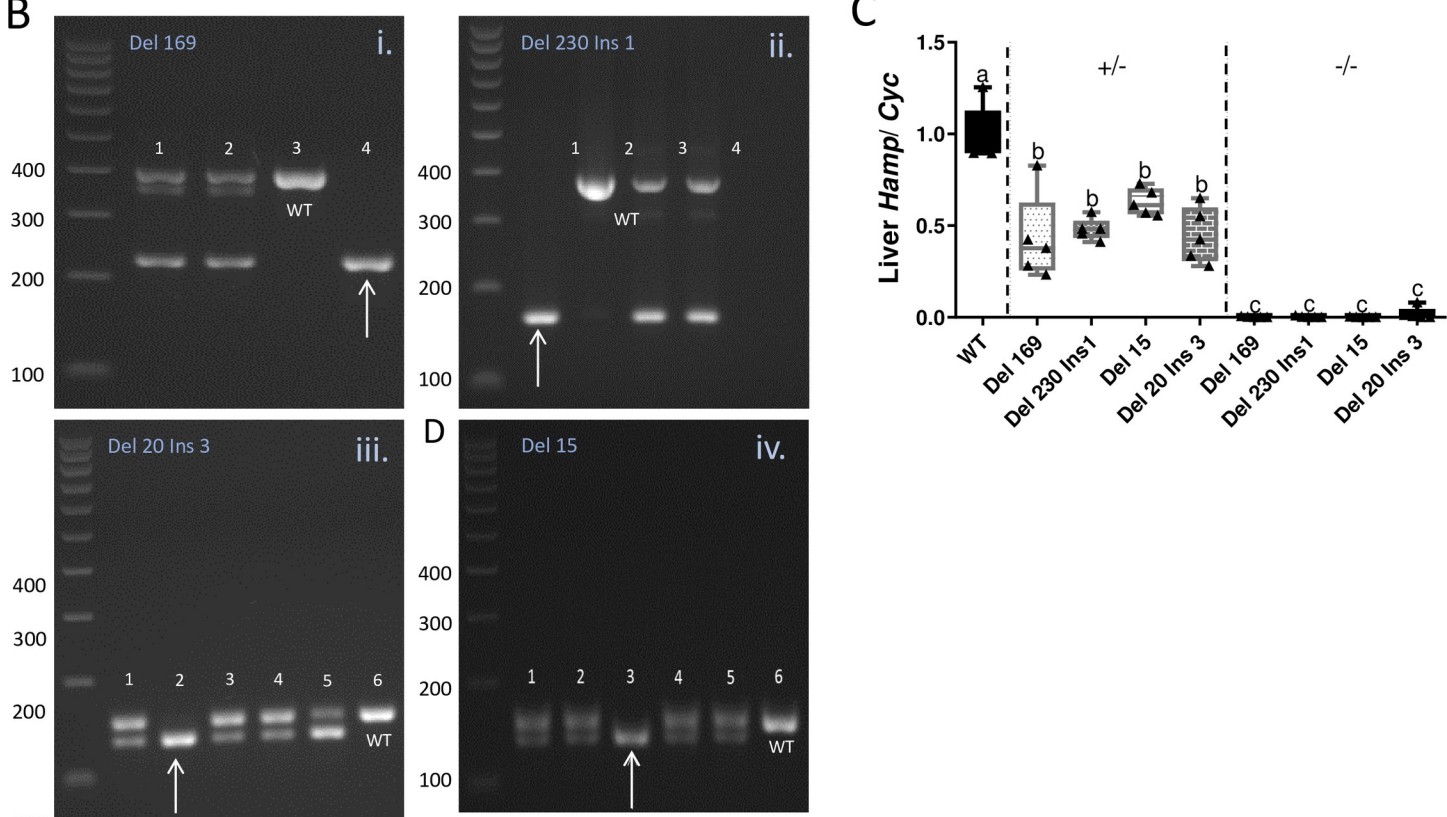

**Fig 1. *Hamp* ablation strategy and confirmation.** Four hepcidin KO rat lines were generated using a TALEN approach, targeting the second exon of the rat *Hamp* gene. The amino acid sequence of rat hepcidin is shown, along with the regions that were deleted in the different mutant rat lines (**A**). PCR genotyping confirmed the appropriate *Hamp* deletions in each KO line (**B**). Representative genomic PCR reactions were loaded on a 2% agarose gel and electrophoresed for 60 min at 120V. A single larger band is indicative of the WT allele; two bands indicate heterozygotes, and one smaller band indicates the homozygous null KOs (see arrows). Expected band sizes were 228 bp and 167 bp for the larger deletions (Del 169 and Del 230 Ins 1, respectively), and 136 bp and 138 bp for the smaller deletions (Del 20 Ins 3 and Del 15, respectively). Bands indicating WT rats were 397 bp for the two larger deletions and 153 bp for two smaller deletions. A 100 bp DNA ladder was used (leftmost lane) for reference. Furthermore, hepatic hepcidin mRNA levels were quantified in WT, heterozygous and homozygous null animals using quantitative RT-PCR (**C**). Data are presented as box plots for n = 5 rats per group and were analyzed by 2-way ANOVA followed by Tukey's multiple comparisons post-hoc test. Means without a common letter differ significantly ($p<0.05$). Mutation × Genotype, p<0.0001.

kidney, heart or pancreas in any of the mutant rats at this age (data not shown). Moreover, heterozygosity for all four *Hamp* mutations did not cause significant hepatic iron loading or splenic iron depletion (Fig 2); therefore, heterozygous animals were excluded from further experimentation.

Since we did not notice any phenotypical differences among the four *Hamp*[-/-] mutant lines, we chose one line for further experimentation. Rats harboring the Del 230 Ins 1 mutation were selected since we surmised that the more sizeable gene interruption (i.e., spanning exons 2 and 3, and causing a frameshift mutation) was most likely to eliminate hepcidin production. For the purposes of this manuscript, *Hamp*[-/-] hereafter refers only to the Del 230 Ins 1 mutant line. To expedite the production of control and experimental animals for further analysis, we bred WT × WT and *Hamp*[-/-] × *Hamp*[-/-]. Importantly, *Hamp*[-/-] breeders exhibited normal reproductive rates and had similar litter sizes as WT rats (data not shown), and growth patterns and organ sizes of offspring from KO breeders did not differ from offspring of WT breeders up to one year of age (Fig 3).

## Ablation of *Hamp* in rats led to iron overload

Serum nonheme iron content was higher in male KO rats at weaning (relative to WT controls), and the difference between genotypes continued to increase up to one year of age (Fig 4A). A similar trend was noted in female KOs, but a significant difference between genotypes was not observed until 9 weeks of age (Fig 4C). As expected, TSAT values paralleled the increases in serum iron in both sexes of *Hamp* KO rats (Fig 4B and 4D).

Blood hemoglobin and hematocrit levels, and other parameters measured by CBC analysis, did not vary between WT and KO rats at any of these ages (data not shown). We next assessed nonheme iron levels in various tissues, first during the post-weaning growth period (i.e., 3 to 9 weeks of age), when iron demands are high. In KO males, hepatic iron loading seemingly began shortly after weaning; however, statistical significance between groups was not achieved

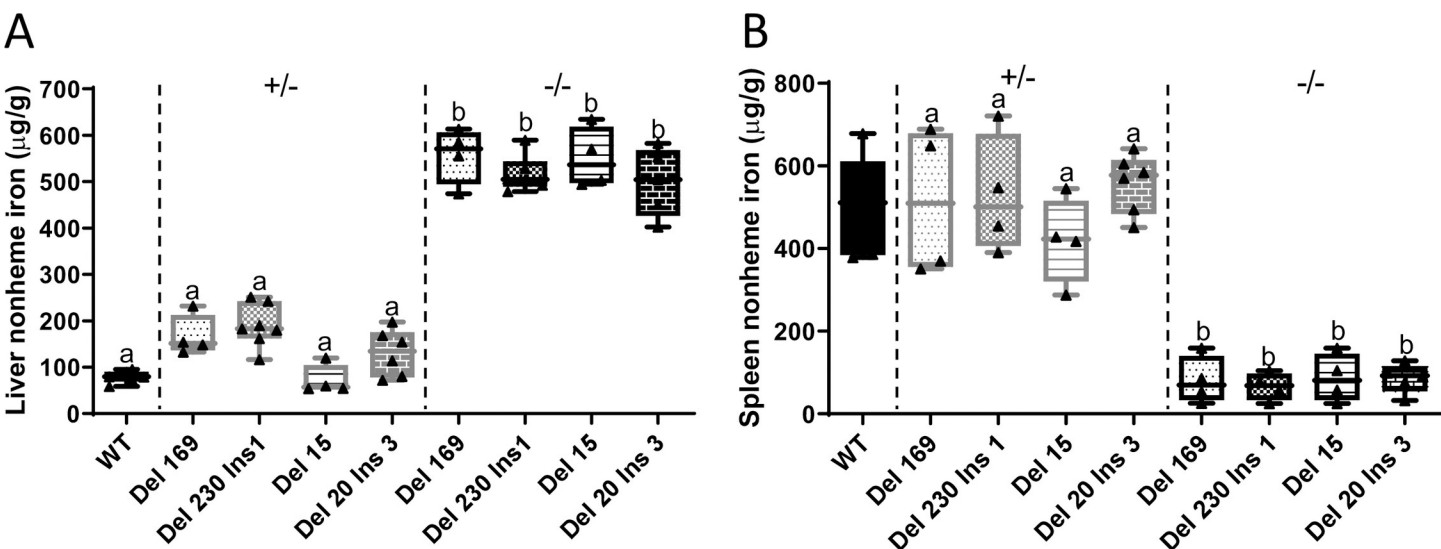

**Fig 2. Homozygosity for four different *Hamp* mutations leads to hepatic iron loading and splenic iron depletion.** Hepatic (**A**) and splenic (**B**) nonheme iron concentrations were determined in tissues derived from male WT (as a control), *Hamp*[+/-] and *Hamp*[-/-] rats at 8–9 weeks of age. Data are presented as box plots for n = 4–6 rats/group. Data were log-transformed to obtain equal variances and subsequently analyzed by one-way ANOVA followed by Tukey's multiple comparisons test; however, untransformed data are presented for ease of interpretation. Groups labeled with different letters vary significantly ($p \leq 0.0129$ for data shown in panel **A**; $p < 0.0001$ for panel **B**). Del, deletion; Ins, insertion; numbers indicate how many base pairs were deleted or inserted.

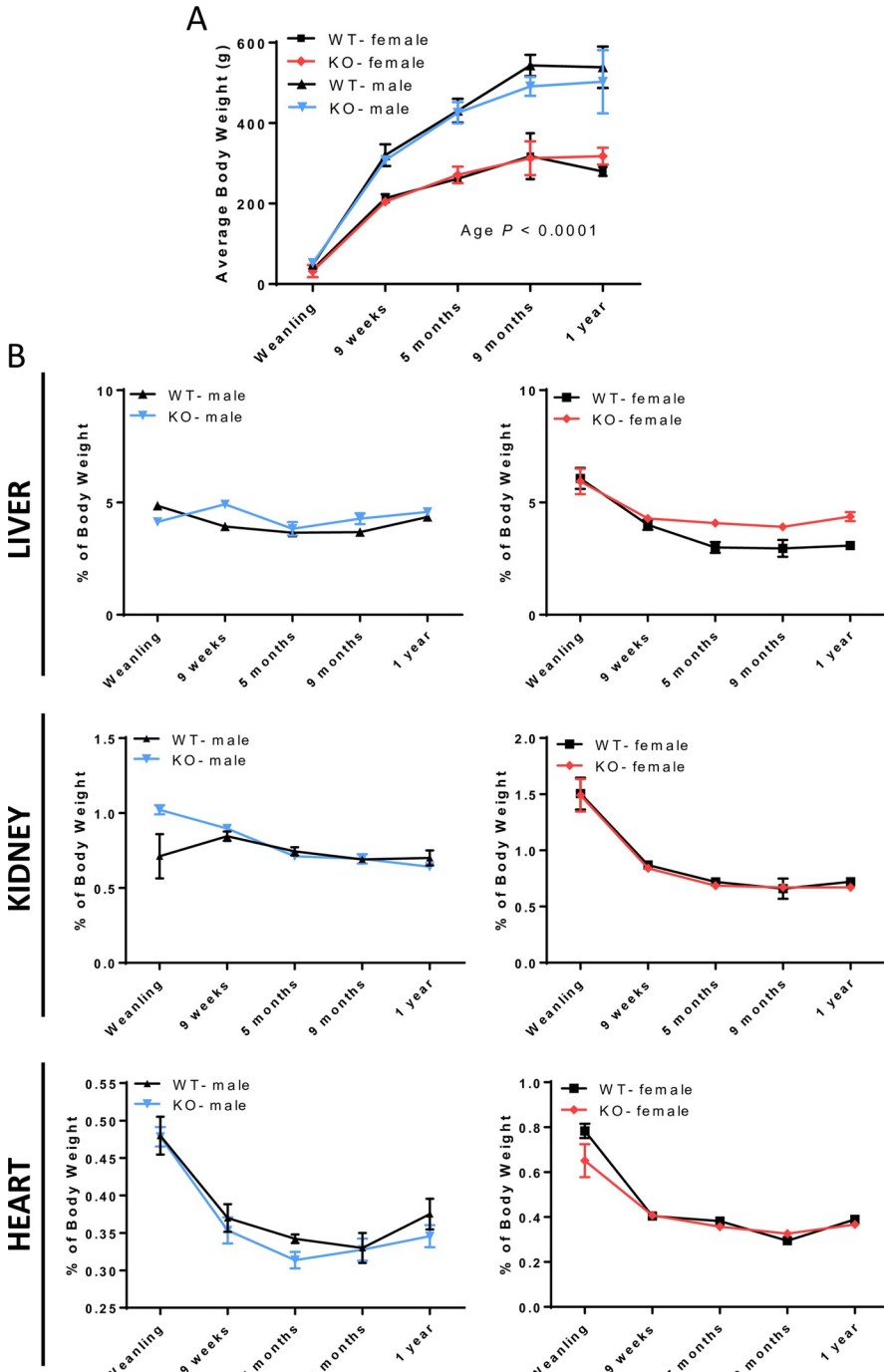

**Fig 3. *Hamp* ablation did not alter growth rates or organ weights in male and female rats.** Average body weights (recorded on the day of sacrifice) of male and female rats are depicted at several different ages (**A**). Data from males and females were analyzed separately but plotted together on the same graph. Organ weights as a percentage of body weight are shown in panel **B**. Data points are means ± SD for n = 8–14 rats/group and represent two independent trials at each age (**A, B**). Data were analyzed by 2-way ANOVA. No significant two-way interactions or main effects were noted (except for an Age main effect for body weight shown in panel **A**).

until 6 weeks of age (Fig 5A). Splenic iron depletion was significant starting at 4.5 weeks of age (Fig 5B). Iron accumulation did not occur within this time frame in pancreas (Fig 5C), heart

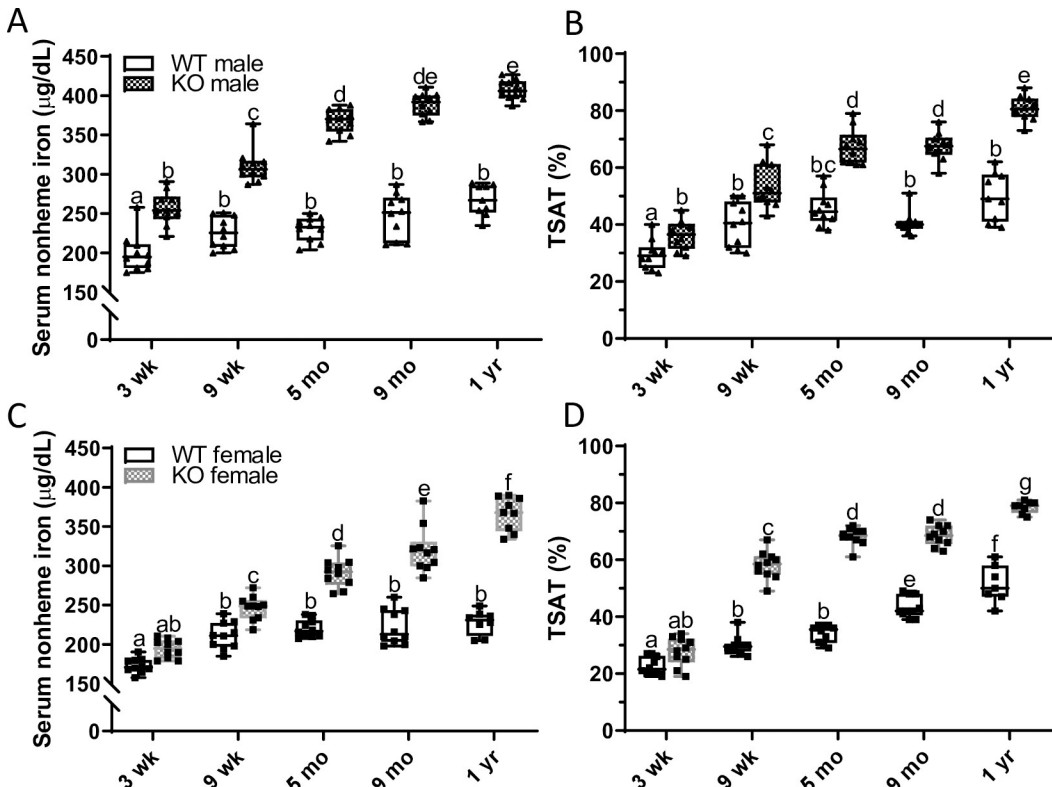

**Fig 4. Lack of hepcidin leads to progressive increases in serum nonheme iron and TSAT in male and female rats.** Serum nonheme iron (**A**, **C**) and TSAT (**B**, **D**) were quantified at different ages in WT and *Hamp*$^{-/-}$ rats. Data are presented as box plots for n = 9–10 rats/group. Data were log transformed before being analyzed by 2-way ANOVA followed by Tukey's multiple comparisons test (**A-D**); however, untransformed data are presented for ease of interpretation. Groups without a common letter differ significantly ($p<0.045$). Genotype × Age, $p<0.0002$; Genotype, $p<0.0001$; Age, $p<0.0001$ (**A-D**).

(Fig 5D), or kidney (Fig 5E). Furthermore, almost identical patterns of tissue iron loading were observed in female *Hamp* KO rats (Fig 6).

Additional longitudinal studies demonstrated progressive hepatic iron loading in male *Hamp* KO rats after 9 weeks of age, being most pronounced at one year of age, when iron levels were ∼15-fold higher than in age-matched, WT controls (Fig 7A). Also, at one year of age, fibrosis was noted in some liver sections, as exemplified in Fig 7B. Perls' Prussian blue staining of liver sections revealed periportal iron accumulation, being most pronounced at one year of age (Fig 7C). A similar pattern of progressive hepatic iron loading was also documented in female *Hamp*$^{-/-}$ rats (Fig 8A).

Furthermore, in male KOs, significant iron accumulation was first noted in the pancreas at 9 months of age, and it peaked at one year of age, when iron levels were >10-fold higher than age-matched WT controls (Fig 9A). Iron deposits were present in exocrine (i.e., acini) and endocrine (i.e., islets) cells of 9-month-old and one-year-old KOs (Fig 9D). Again, a similar chronology of progressive pancreatic iron loading was also documented in female *Hamp*$^{-/-}$ rats (Fig 8B). In kidney, iron accumulation was noted in male KO rats beginning at 9 months of age (Fig 9B), while in heart, iron loading was significant starting at 5 months of age (Fig 9C). Similar results were also documented in female *Hamp*$^{-/-}$ rats (Fig 8C/8D). Iron deposits were observed mainly in the renal cortex (Fig 9E), and throughout the cardiac muscle fibers (Fig 9F).

Splenic nonheme iron depletion was observed in male (Fig 10A) and female (Fig 8E) KO rats up to one year of age. In some older KOs, the spleen had an abnormal appearance, being

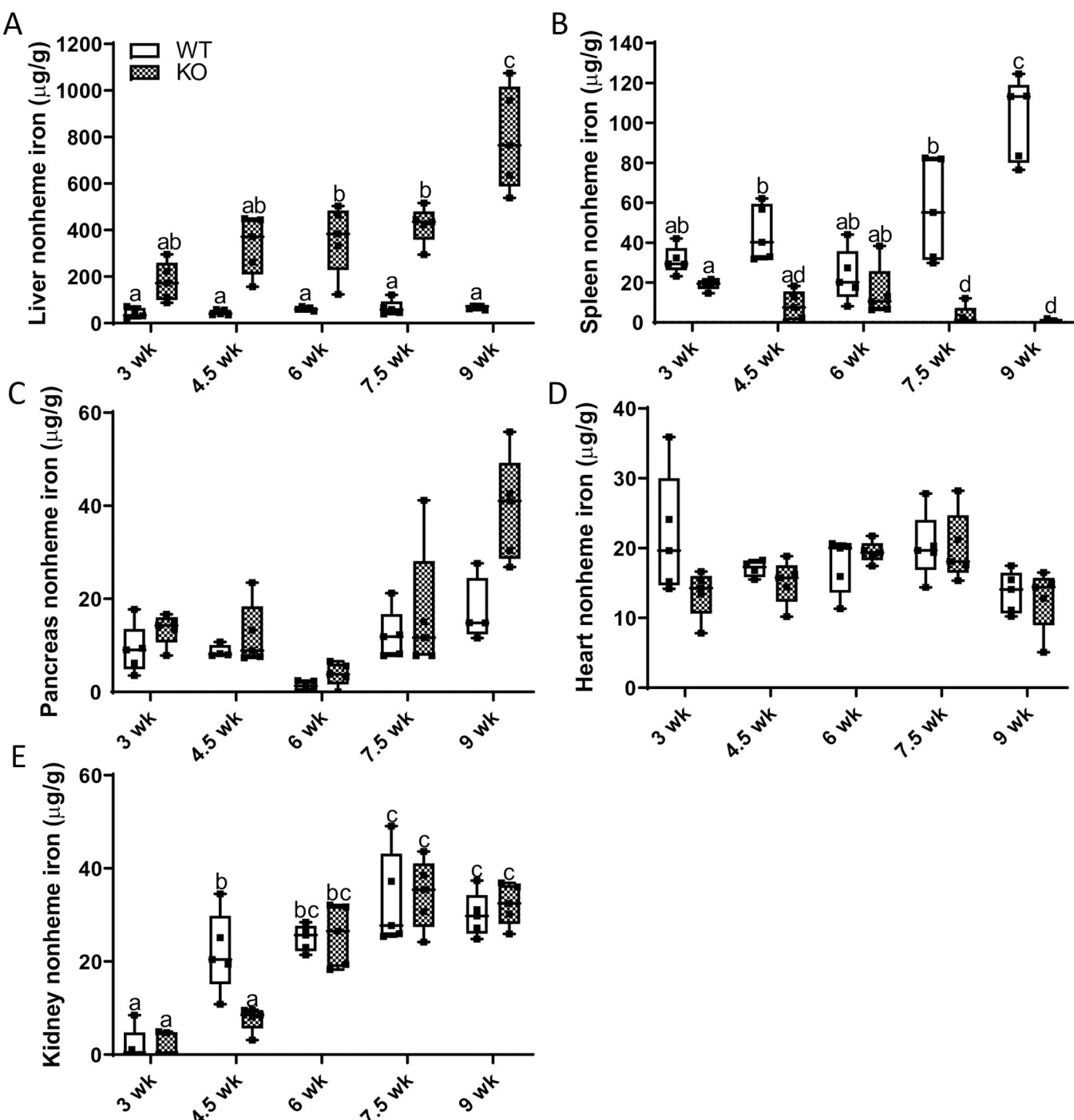

**Fig 5. Hepatic iron loading and splenic iron depletion begins shortly after weaning in male *Hamp* KO rats.** To determine the chronology of iron loading, tissue nonheme iron concentrations were assessed every week and a half beginning at weaning in rats of both genotypes. Shown are nonheme iron levels in the liver (**A**), spleen (**B**), pancreas (**C**), heart (**D**) and kidney (**E**). Data are presented as box plots for n = 4–5 rats per group. All data were log transformed before being analyzed by 2-way ANOVA followed by Tukey's multiple comparisons test; however, the untransformed data are presented for ease of interpretation. Groups without a common letter differ significantly ($p \leq 0.014$). Genotype × Age, $p \leq 0.019$ (**A, B, E**); Genotype, $p < 0.0001$ (**A, B**); Age, $p \leq 0.016$ (**A-E**).

brown (instead of dark red) and having a metallic sheen (Fig 10C). This splenic abnormality was documented in 65% of males (n = 32) that were sacrificed at > 9 months of age, and 13% of

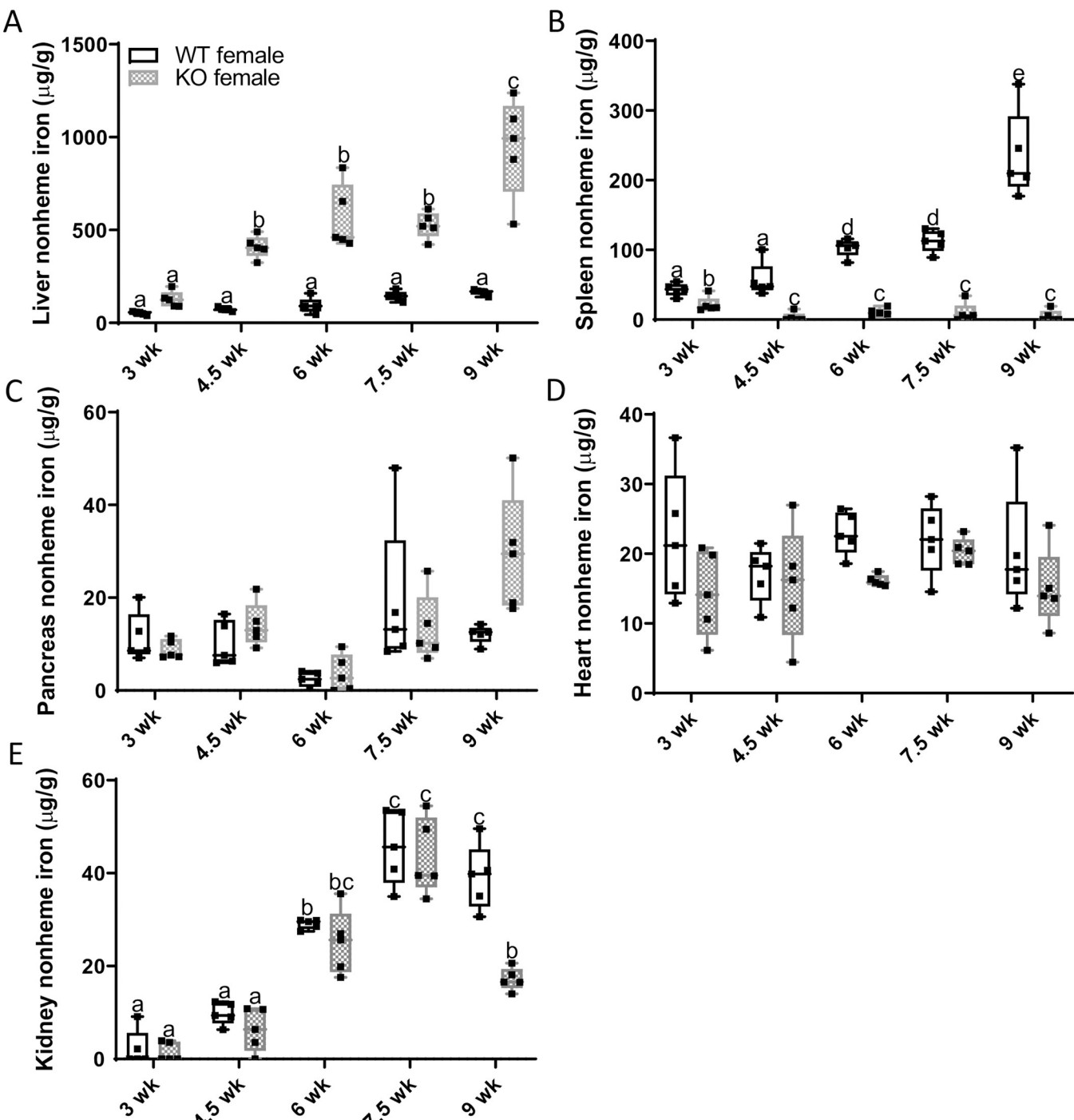

**Fig 6. Hepatic iron loading and splenic iron depletion begins soon after weaning in female *Hamp* KO rats.** To determine the age of initial iron loading, tissue nonheme iron levels were quantified in female *Hamp*$^{-/-}$ rats and WT littermates at various ages after weaning. Shown are nonheme iron levels in the liver (**A**), spleen (**B**), pancreas, (**C**), heart (**D**) and kidney (**E**). Data are presented as box plots for n = 4–5 rats/group and were analyzed by 2-way ANOVA followed by Tukey's multiple comparisons test. Some data were first log-transformed due to unequal variance (**C-E**); however, untransformed data are presented for ease of interpretation. Groups labeled with different letters vary significantly (p<0.041; **A, B, E**). Genotype × Age, $p \leq 0.0003$ (**A, B, E**); Genotype, $p \leq 0.0093$ (A, B, D, E); Age, $p \leq 0.008$ (**A-C, E**).

older females (n = 30), although the severity was much less in females (i.e., the spleens only had one or a few brown spots). The relative red/white pulp ratio was also altered in the abnormal

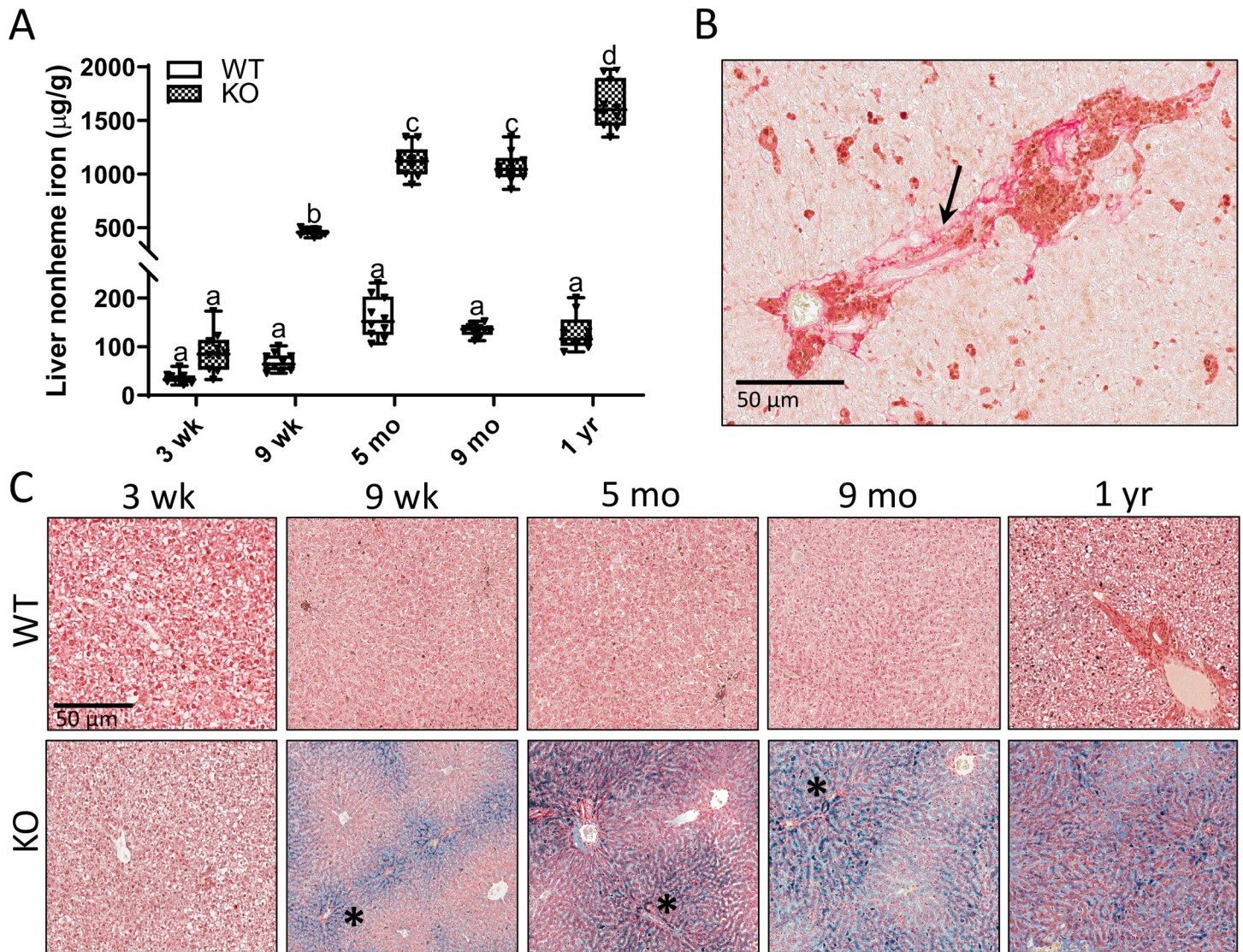

**Fig 7. Progressive periportal iron loading is observed in male *Hamp*$^{-/-}$ rats.** Hepatic nonheme iron levels in WT and *Hamp*$^{-/-}$ rats are shown in panel **A**. Data are presented as box plots for n = 9–10 animals per group. Data were log transformed before being analyzed by 2-way ANOVA followed by Tukey's multiple comparisons test; however, untransformed data are presented for ease of interpretation. Groups without a common letter differ significantly ($p<0.0001$). Genotype × Age, $p<0.0001$; Genotype, Age, $p<0.0001$. An example picro-sirius red staining image of a liver section from a 1-year-old male *Hamp*$^{-/-}$ rat is also shown (**B**). The arrow indicates a fibrotic region (i.e., collagen fibers). Fibrosis was infrequently observed in the livers of older KO rats. Representative images of liver sections from WT and *Hamp*$^{-/-}$ rats stained with Perls' Prussian blue are also shown (**C**) (n = 6 per group). Asterisks demarcate portal triads.

spleens from KO males, with a relative expansion of the red pulp and a corresponding diminution of the white pulp (Fig 10B). Otherwise, these brown spleens were equivalent in length and weight to the (normal appearing) spleens from WT and unaffected *Hamp*$^{-/-}$ rats. Furthermore, iron staining was apparent in spleens from WT rats starting at 9 weeks of age and it peaked at one year, while iron deposits were less prevalent in spleens from KOs (Fig 10D).

## Acute DSS exposure caused localized colonic inflammation

Food and water intake were largely unaffected by DSS treatment in both sexes and genotypes of rats (Fig 11). Pre- and post-mortem assessment of biomarkers of disease activity

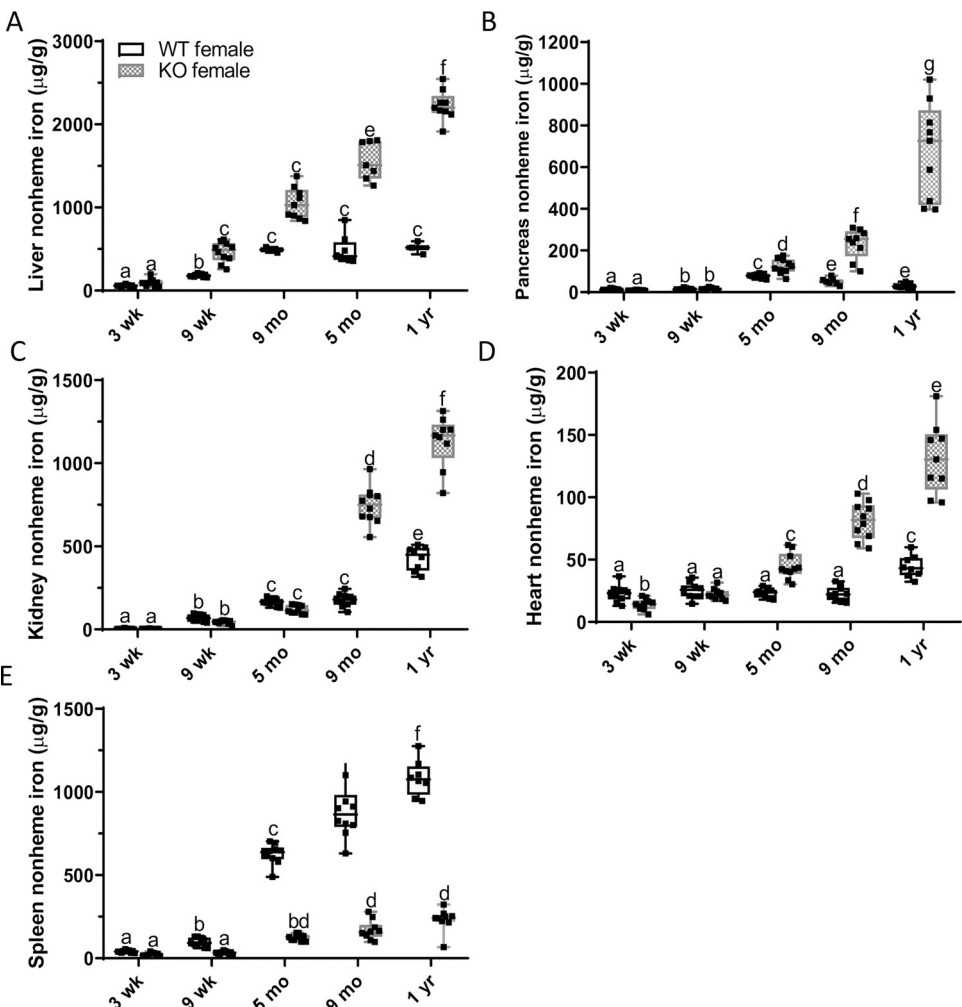

**Fig 8. Multi-visceral iron loading and spleen iron depletion over time in female rats.** To characterize the chronology of iron loading, tissue nonheme iron concentrations were determined at different ages in female WT and *Hamp*^-/- rats. Shown are nonheme iron levels in the liver (**A**), pancreas (**B**), kidney (**C**), heart (**D**) and spleen (**E**). Data are presented as box plots for n = 9–10 rats per group and were analyzed by 2-way ANOVA followed by Tukey's multiple comparisons test (**C-E**). Some data were first log-transformed due to unequal variance (**C-E**); however, untransformed data are presented for ease of interpretation. Groups labeled with different letters vary significantly ($p \leq 0.0274$). Genotype × Age, $p \leq 0.0001$ (**A-E**); Genotype, $p \leq 0.0001$, Age, $p \leq 0.0001$ (**A-E**).

was utilized to determine the pathophysiological impact of the acute DSS-exposure protocol [20] Male rats treated with DSS had decreased colon lengths, but no difference was noted in females (Fig 12A). DAI scores were essentially zero in untreated (control) animals, as expected; however, a progressive increase in DAI was observed in DSS-treated male and female WT rats (Fig 12B). These scores mainly related to changes in stool consistency and the presence (or absence) of hemoccult since weight loss was minimal. Intestinal pathology was observed in both sexes, but outcomes were more severe in males. Development of colitis was further demonstrated by the significantly higher macroscopic histology scores from colonic sections from DSS-treated WT male rats (Fig 12C) and by examination of H&E-stained tissue sections (Fig 13). Additionally, we further sought to determine whether DSS treatment caused barrier breach and concomitant systemic inflammation. This was accomplished by quantifying serum pro-inflammatory cytokines by ELISA (for IL-6) and using a

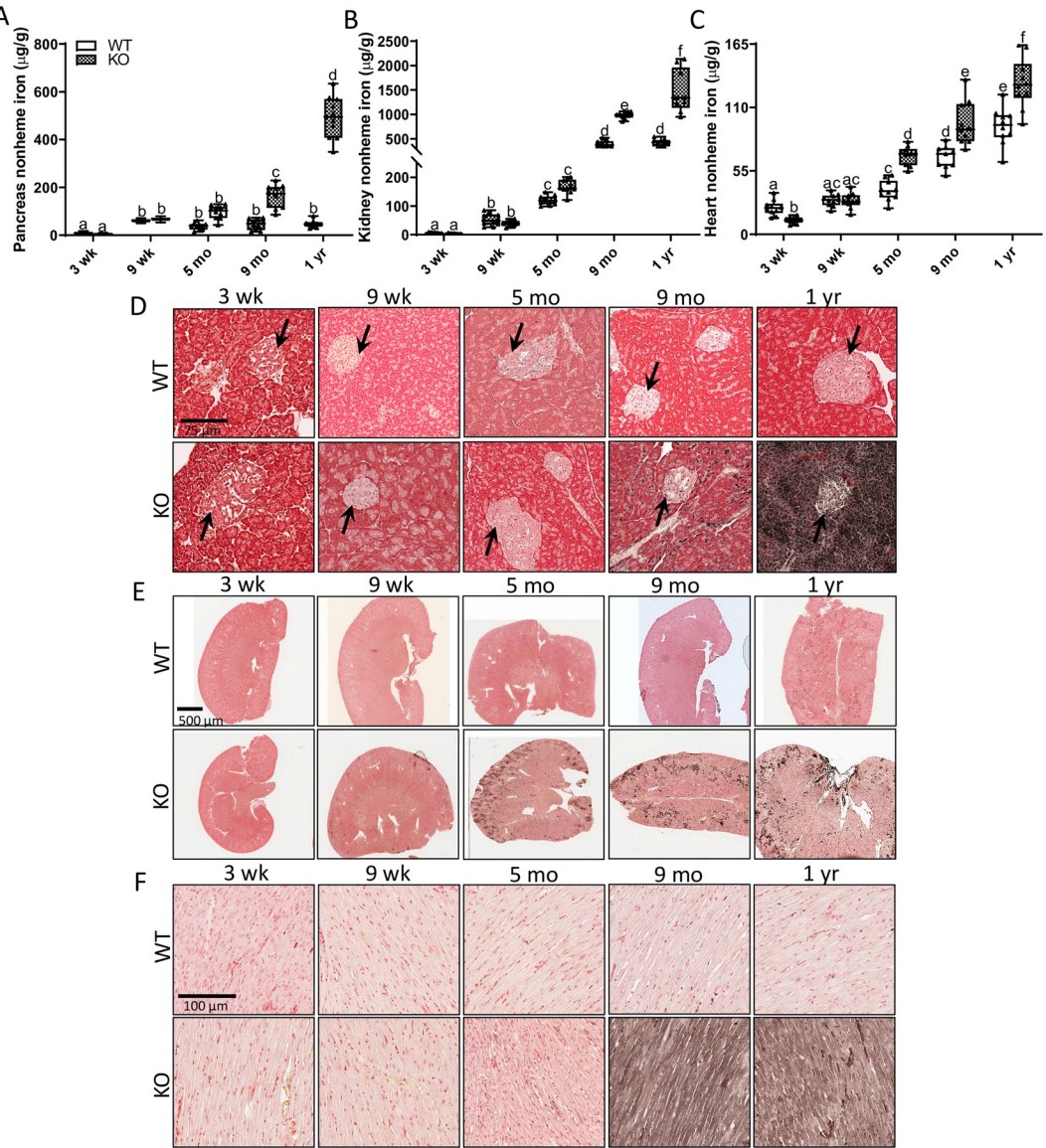

**Fig 9. Male *Hamp*⁻/⁻ rats display progressive iron loading in extra-hepatic tissues.** To characterize the chronology of iron loading in various tissues, nonheme iron concentrations were quantified and Perls' Prussian blue iron staining was performed in samples collected from rats of both genotypes at different ages. Nonheme iron levels in the pancreas, kidney and heart are shown in panels **A-C**. Data are presented as box plots for n = 9–10 rats/group. All data were log transformed before being analyzed by 2-way ANOVA followed by Tukey's multiple comparisons test; however, untransformed data are presented for ease of interpretation. Groups without a common letter differ significantly ($p < 0.043$). Genotype × Age, $p < 0.0007$; Genotype, $p < 0.022$; Age, $p < 0.001$ (**A-C**). Representative images are also shown from iron-staining experiments using paraffin-embedded sections derived from the pancreas (**D**), kidney (**E**), and heart (**F**) of male WT and *Hamp*⁻/⁻ rats at various ages (n = 6 per group). In panel **D**, arrows indicate pancreatic islets (note iron deposits within the islets of older KOs).

rat cytokine array. Both experimental approaches showed no changes in serum cytokine levels during acute colitis (Figs 12D and 14). These results indicate that our DSS treatment protocol caused localized intestinal inflammation, but did not induce a more severe, systemic inflammatory response.

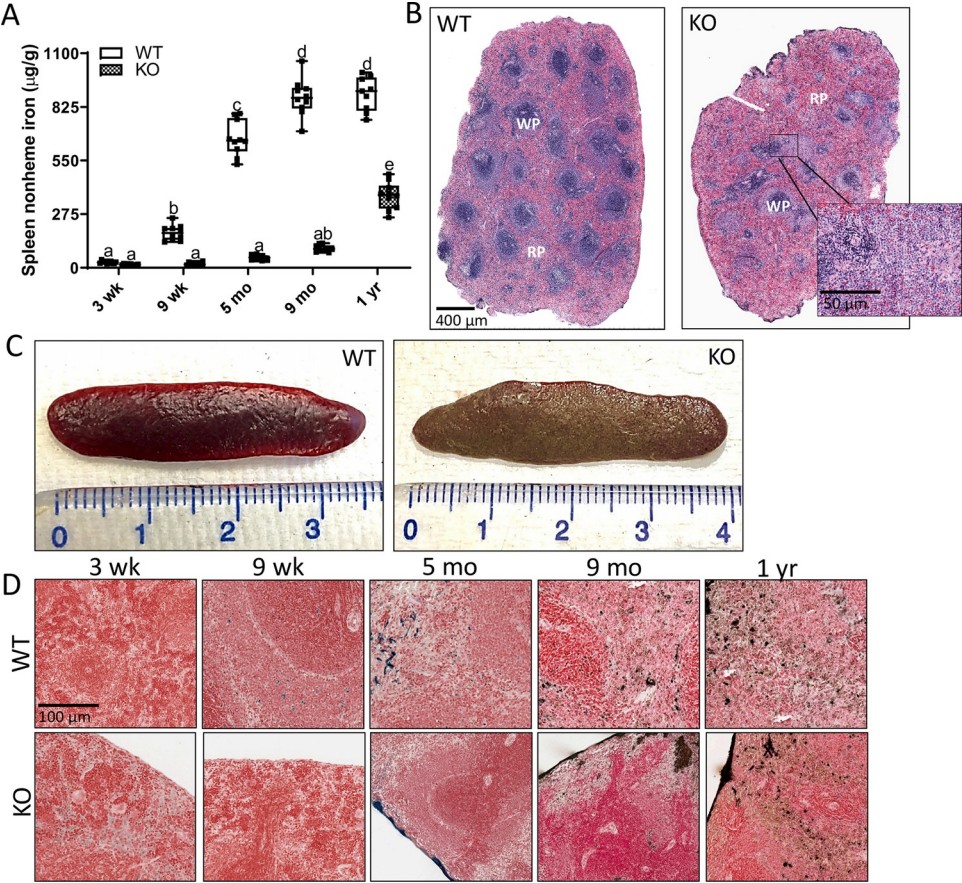

**Fig 10. Splenic iron is depleted in male *Hamp*[-/-] rats.** Nonheme iron levels were assessed in spleens of *Hamp* KO rats at various ages (**A**). Data are presented as box plots for n = 8–10 rats/group. Data were log transformed before being analyzed by 2-way ANOVA followed by Tukey's multiple comparisons test; however, untransformed data are presented for ease of interpretation. Groups without a common letter differ significantly ($p \leq 0.048$). Genotype × Age, $p < 0.0001$; Age, Genotype $p < 0.0001$. Representative images of H&E staining of paraffin-embedded spleen sections from 10-month-old male WT and *Hamp*[-/-] rats are also depicted (**B**). In ~65% of older male KOs (i.e., >9 months of age), the spleens had an abnormal appearance, and the relative red pulp (RP)/white pulp (WP) ratio was altered (compared to normal spleens from unaffected KOs and WT rats) (**C**) ($p < 0.05$ by 1-way ANOVA) [n = 5/group]. The ruler indicates centimeters. Representative images of Cobalt DAB-enhanced Perls' Prussian blue iron staining in paraffin-embedded spleen sections at different ages are also shown (**D**).

## DSS treatment disrupted iron homeostasis in WT rats

We next sought to determine whether oral DSS exposure altered systemic iron homeostasis. Bioindicators of iron status were thus quantified in both sexes of WT control and DSS-treated rats. Blood Hb levels and serum nonheme iron decreased in male rats during acute colitis, while no change in these parameters was observed in females (Fig 15A and 15B). TSAT was lower in DSS-treated rats of both sexes, but the decrease was of larger magnitude in males (Fig 15C). In males, liver nonheme iron levels were unaffected by acute colitis, while unexpectedly, a huge increase was seen in females (Fig 15D). The (patho)physiological significance of this increase in hepatic storage iron in females is unclear. Moreover, spleen iron stores were depleted in rats of both sexes during intestinal inflammation, but again, the magnitude of decrease was more significant in males (Fig 15E). And lastly, serum ferritin (Fig 15F) and liver hepcidin mRNA expression (Fig 15G) did not vary by sex or treatment. Collectively, these experimental observations have two important implications: first, liver *Hamp* expression and serum ferritin levels

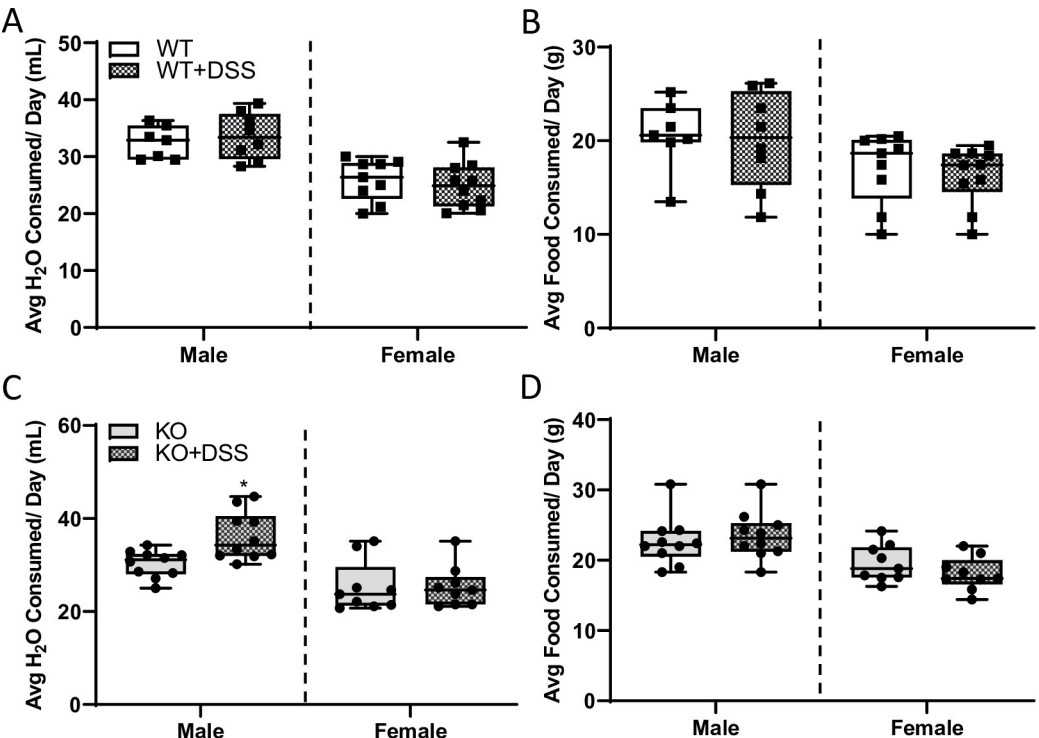

**Fig 11. Food and water intake did not differ between control and DSS-treated male or female rats of the same genotype.** Water intake (**A**, **C**) and food consumption (**B**, **D**) were measured daily in rats of both sexes for the duration of the DSS treatment-period including at baseline Day 0 before treatment began. Data are presented as box plots for two independent experiments with n = 7–11 rats/group and were analyzed by 2-way ANOVA followed by Tukey's multiple comparisons post hoc test. No significant two-way interactions were noted. Genotype $p \leq 0.05$ (**B, D**); Treatment, $p = 0.027$ (**C**).

were unaffected by DSS treatment, indicating a lack of systemic inflammation (since both are acute-phase reactants that increase during infection and inflammation); and second, increased erythropoietic demand associated with anemia, and depletion of serum iron and splenic iron stores that occurred during acute colitis, did not suppress hepatic *Hamp* expression (as anticipated). Importantly, previous studies have demonstrated that hepatic hepcidin mRNA levels correlate strongly with circulating levels of the functional hormone [29], which is logical since hepcidin production is predominantly regulated at the transcriptional level [30].

### Intestinal iron absorption increased during acute colitis

Absorption of iron ($^{59}$Fe) from an orally administered test dose was quantified by dividing the radioactive counts in a rat immediately after dosing by the radioactivity remaining in the rat 24 hours later (just prior to euthanasia and necropsy). By these measures, localized intestinal inflammation stimulated iron absorption in WT rats of both sexes (Fig 16A and 16I), but despite this, $^{59}$Fe in the blood was significantly lower (Fig 16B). In males, $^{59}$Fe accumulated in the liver, while there was no apparent increase in females (Fig 16C). In other tissues, including kidney, bone, muscle, and spleen, $^{59}$Fe activity was markedly lower in males and females upon DSS exposure (Fig 16D–16G). In the heart, $^{59}$Fe accumulation was drastically reduced in DSS-treated males, while no change was observed in females (Fig 16H). In sum, consistent with the noted anemia and decreases in serum nonheme iron (in males) and reduction in TSAT and depletion of splenic iron stores (in both sexes) (see Fig 15), intestinal inflammation increased assimilation of an $^{59}$Fe test dose. Surprisingly,

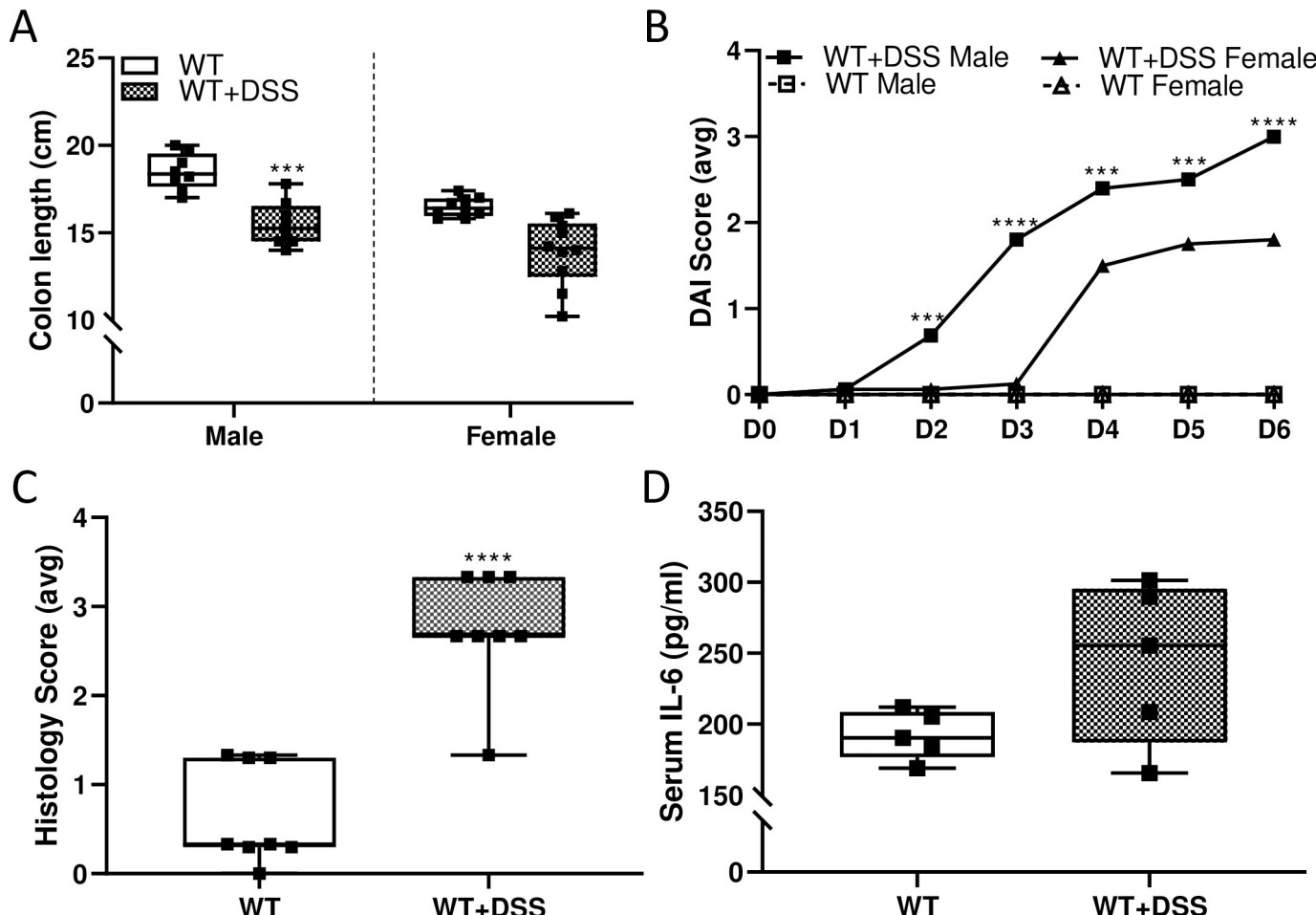

**Fig 12. Oral DSS administration caused colitis with GI tract-restricted inflammation in 10-week-old rats of both sexes.** Length of the colon, defined as the end of the cecum to the anus, was measured as a macroscopic indicator of DSS-related colonic damage (**A**). Data are presented as box plots for n = 7–11 rats/group and were analyzed separately by sex using a two-sample t test (***$p<0.001$). Disease Activity Index (DAI) scores were also calculated daily (**B**). Average scores on each day for each sex and experimental group are depicted in the graph. Data for WT+DSS male and WT+DSS female were compared by two-way ANOVA with repeated measures (Day × Group, $p<0.001$). Individual t-tests were then used to compare groups at each treatment day (***$p<0.001$; ****$p<0.0001$); n = 7–11 rats/group. Furthermore, H&E-stained tissue sections from the descending colon of male WT rats were blindly scored in 3 randomly selected areas, and histology scores were generated (**C**). Data are presented as box plots for n = 10 rats/group and were analyzed by a two-sample t test (****$p<0.0001$). Serum IL-6 levels from male rats were also assessed by ELISA (**D**). No significant difference was noted between groups (by two-sample t test).

however, freshly absorbed iron was not retained in the blood (where it could increase TSAT) nor did it appear to be preferentially destined for the bone marrow (where it could support enhanced erythropoiesis). Increased iron accumulation in the liver (at least in males) could reflect increased iron requirements to produce immune cells (e.g., Tregs) which suppress the immune response associated with gut inflammation (and which have been shown to increase in abundance in DSS colitis) [31].

## DSS-colitis is associated with GI tract-restricted inflammation in *Hamp* KO rats

Short-term DSS-exposure caused colitis in iron-loaded, *Hamp* KO rats, as exemplified by shortening of the colons (Fig 17A), progressively increasing DAI scores (Fig 17B), and higher colonic histology scores (Fig 17C) and macroscopic damage (Fig 13C and 13D). Disease severity was

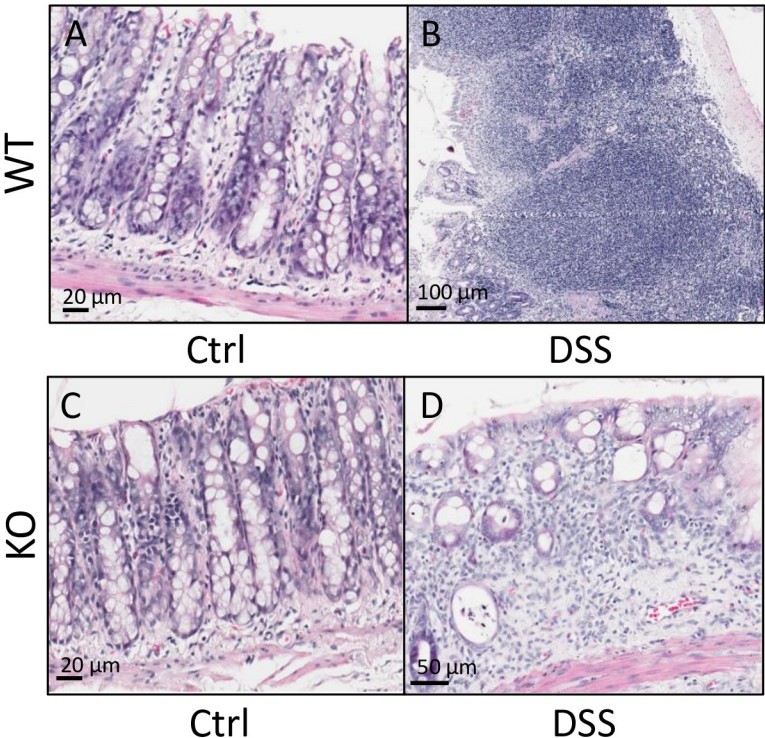

**Fig 13. DSS exposure causes colonic damage in WT and *Hamp* KO rats.** Rats were euthanized at the conclusion of the treatment course and at necropsy, dissected colonic sections were processed into Swiss Rolls to allow for microscopic analysis. Representative H&E-stained sections from wildtype males illustrate immune cell infiltration, and structural damage in the distal colon with DSS treatment (**A**, **B**). Similar pathological changes were observed in KO rats (**C**, **D**).

enhanced in males, as evidenced by more significant colon shortening, and higher DAI scores from day 3 through day 6. Food intake did not differ between experimental groups, and only a minor (but probably biologically insignificant) increase in water intake was noted in male *Hamp* KOs in the DSS group (Fig 11C and 11D). Furthermore, systemic inflammation did not occur, as evidenced by no increase in serum IL-6 (Fig 17D) or other cytokines assayed using the rat cytokine array (Fig 14). Importantly, DSS exposure caused similar pathophysiological changes in male WT and *Hamp* KO rats (e.g., DAI scores were not statistically different between geno-types); to aid in interpretation, Fig 18 shows direct comparisons between genotypes.

In sum, these data demonstrate that the pathological response to oral DSS exposure is not influenced by *Hamp* ablation or iron accumulation in parenchymal tissues. This result is per-haps not surprising since in our model of acute colitis, minor tissue damage and inflammation are limited to the GI tract, thus obviating a possible influence from systemic inflammation (which would likely exacerbate pathological outcomes). Moreover, the intestinal epithelium in *Hamp*$^{-/-}$ rats is paradoxically iron depleted, rather than iron loaded (which would have predic-tively increased tissue damage due to enhanced oxidative stress). This iron depletion occurs because of high FPN1 iron export activity (due to lack of hepcidin), which creates a functional iron deficit in duodenal enterocytes [32].

## Systemic iron homeostasis is disrupted by acute colitis in *Hamp* KOs

DSS-exposure had different pathophysiological effects in *Hamp* KOs, as compared to WTs rats, and sex differences were also noted. Somewhat surprisingly, acute colitis was associated with

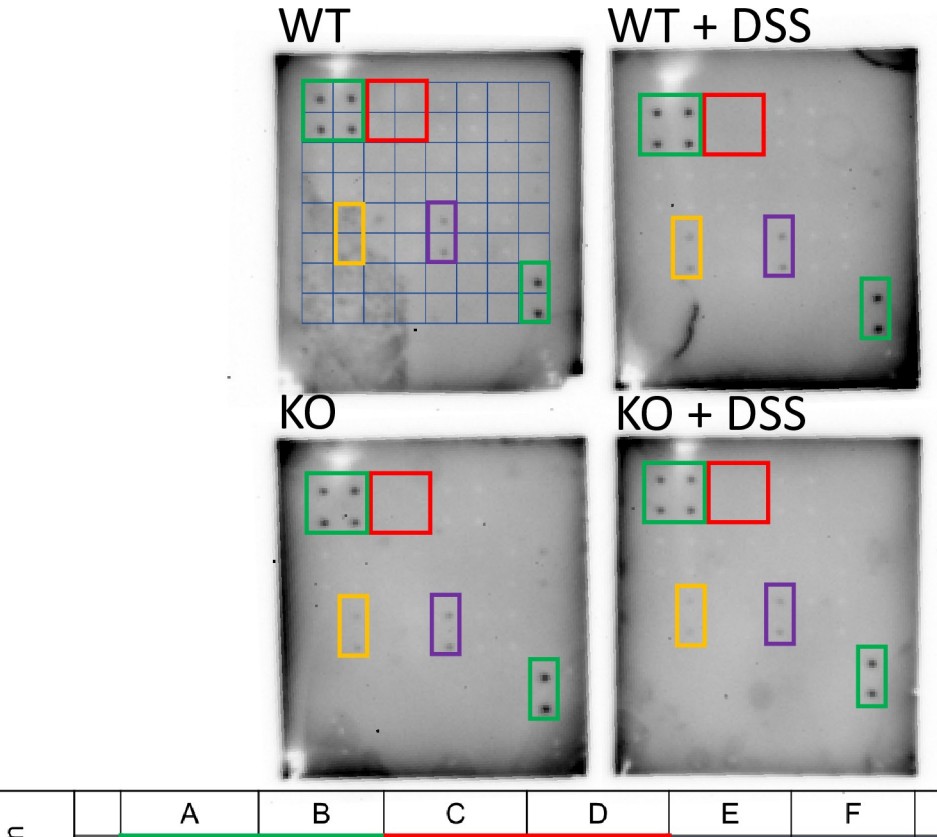

**Fig 14. DSS treatment did not cause systemic inflammation in male rats of either genotype.** Serum cytokines were assayed using a rat cytokine antibody array (AAR-CYT-1; RayBiotech, Norcross, GA). A grid was drawn on one blot for orientation, and positive (green) and negative (red) controls are indicated with boxes. The positive and negative controls worked as expected, validating the assay. The layout of the blots is shown in the table below (as provided by the manufacturer). Only two cytokines were positive MCP-1 (yellow boxes) and TIMP-1 (purple boxes), but levels were not noticeably different between genotypes or treatment groups.

increased blood Hb levels in male KOs (Fig 19A); in fact, Hb levels increased to above values of age-matched, WT controls ($p < 0.0001$; two-way ANOVA with Tukey's multiple comparisons post-hoc test). In contrast, in female KOs, blood Hb levels decreased upon DSS-treatment (Fig 19A). Serum nonheme iron was decreased in both sexes of KO rats during colitis (Fig 19B). TSAT increased in male KOs and decreased (very slightly) in females upon DSS exposure (Fig 19C). Liver and spleen nonheme iron content also varied by sex, with intestinal inflammation decreasing levels in males, but increasing levels in females (Fig 19D and 19E). And lastly, serum ferritin increased in female KO rats during acute colitis (but not in males) (Fig 19F).

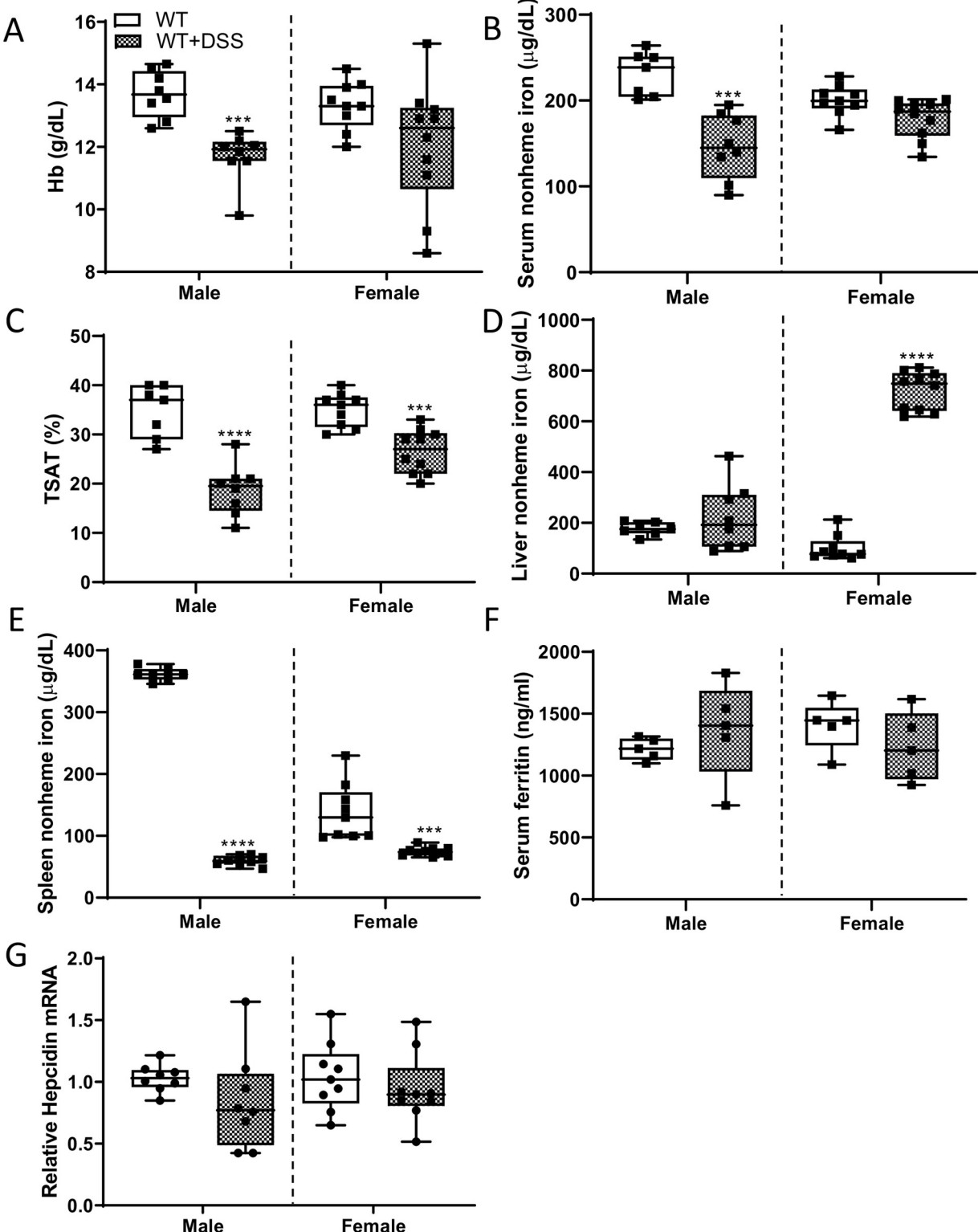

**Fig 15. Acute colitis causes a reduction in serum nonheme iron and TSAT, and depletion of splenic iron stores in WT rats.** Various iron-related physiological parameters were assessed in 10-week-old control and DSS-treated WT rats of both sexes. Hb levels are presented in panel **A**. Also shown are serum nonheme iron levels (**B**), TSAT (**C**), and liver (**D**) and spleen (**E**) nonheme iron levels. ELISA results for serum ferritin (**F**) and liver hepcidin mRNA expression by qRT-PCR (**G**) are also shown. Data are presented as box plots representing two independent experiments with n = 5–11 rats/group with sexes analyzed separately using a t-test (***$p<0.001$; ****$p<0.0001$.

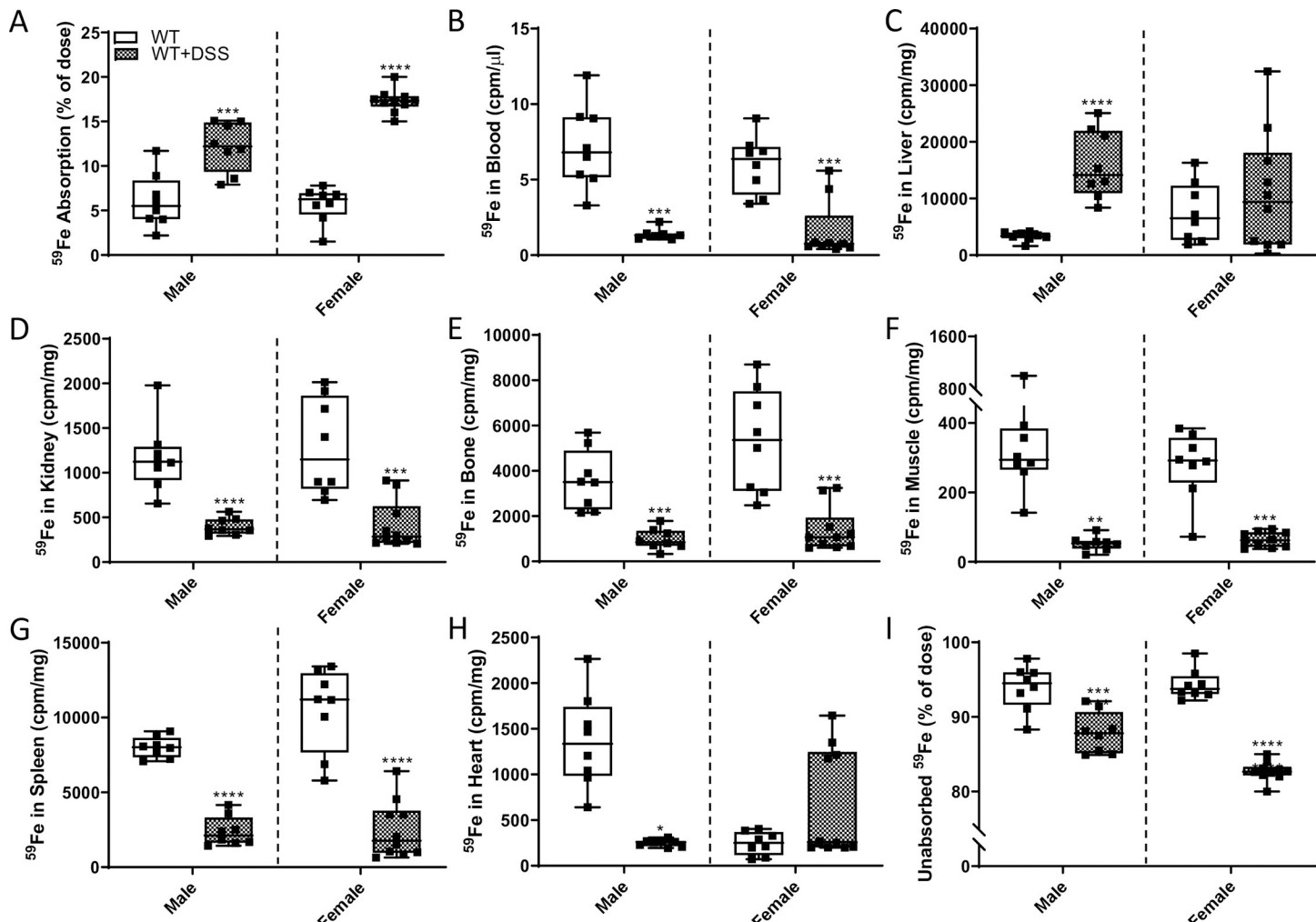

**Fig 16. DSS treatment stimulated intestinal iron absorption in male and female WT rats.** Iron absorption and tissue distribution were assessed in 10-week-old rats of both sexes with and without intestinal inflammation. The absorption of a test dose of $^{59}$Fe administered by oral, intragastric gavage is presented as the percentage of radioactivity present in the carcass (minus $^{59}$Fe trapped in the GI tract) 24 hours after dosing (**A**). Distribution of $^{59}$Fe in several tissues was also assessed by gamma counting (**B-H**). The percentage of unabsorbed $^{59}$Fe is also depicted (**I**). All data are presented as box plots for n = 7–11 rats/group from two independent experiments and were analyzed separately by sex using a t-test ($^*p < 0.0241$; $^{**}p < 0.0035$; $^{***}p < 0.0009$; $^{****}p < 0.0001$).

Furthermore, when comparing to male WTs, serum and liver nonheme iron, and TSAT were elevated in male KO rats, while spleen iron was lower (as would be expected in this genetic iron overload model) (see Figs 20 and 21 for genotype comparisons). Overall, these observations demonstrate that iron homeostasis is differentially influenced by localized intestinal inflammation, depending upon the sex, genotype and resulting phenotype of the animal. Sex differences may reflect the variable disease-related symptoms associated with acute DSS exposure, with more severe GI disease noted in males. Other variations in the pathological response could then relate to extra-hepatic *Hamp* production (in WTs), or increased body iron stores (in KOs).

## Intestinal iron absorption decreases in male *Hamp* KO rats, but is unchanged in females

*Hamp* KO rats have maximal iron stores (as well as pathological tissue iron accumulation), and as such, increased physiological demand for iron could be met by mobilization of storage iron.

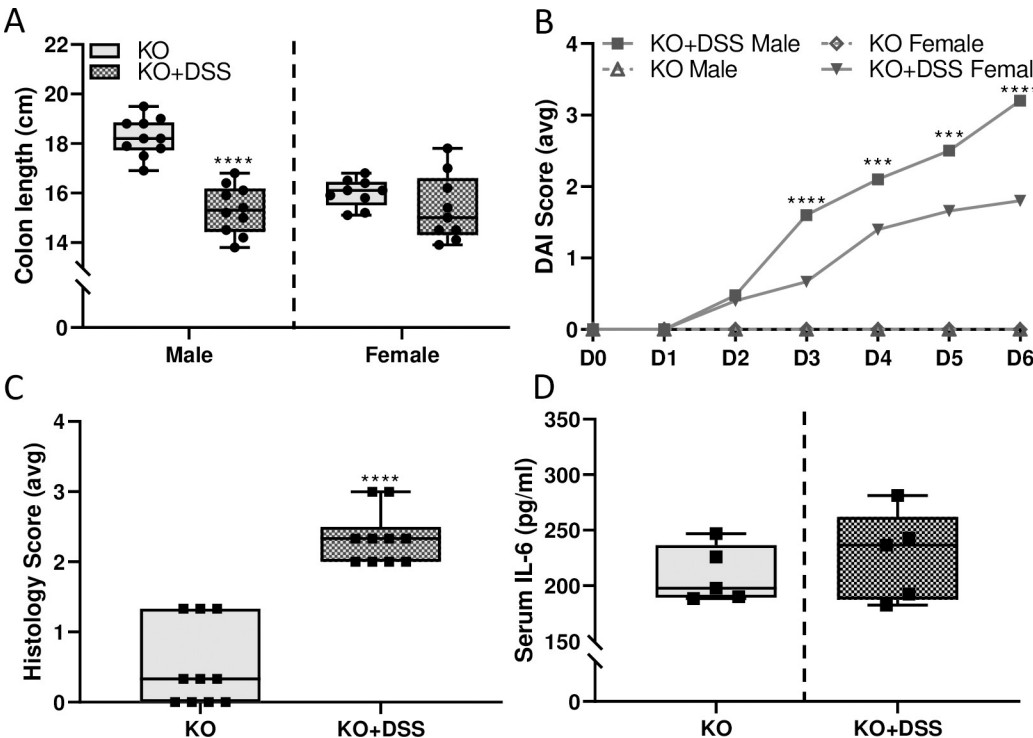

**Fig 17. Acute DSS exposure causes colitis with GI tract-restricted inflammation in *Hamp* KO rats of both sexes.**
Length of the colon, defined as the end of the cecum to the anus, was measured as a macroscopic indicator of DSS-related intestinal damage in 10-week-old male and female *Hamp* KO rats (**A**). Data are presented as box plots for n = 7–11 rats/ group and were analyzed separately by sex using a t-test (****$p < 0.0001$). Disease activity index (DAI) scores were also determined daily for rats of both sexes (**B**). Scores were averaged on each day for each group. Data for KO+DSS male and KO+DSS female were compared by two-way ANOVA with repeated measures (Day x Group, $p < 0.001$). Individual t-tests were then used to compare groups at each treatment day (***$p < 0.001$; ****$p < 0.0001$); mean values are shown for n = 7–11 rats/group. Histology scores from H&E-stained colonic sections from male rats are shown in panel **C**. Sections from the descending colon were blindly scored in 4 randomly selected areas. Each area was assigned a score from 0–4 based on levels of inflammatory cell infiltration, goblet cell depletion, and damage to the crypt/villus architecture. Data are presented as box plots for n = 10 rats/group and were analyzed by t-test (****$p < 0.0001$). Serum IL-6 levels were assessed in male rats by ELISA (**D**). No significant difference (by t test) was noted between controls or DSS-treated rats (indicating a lack of systemic inflammation).

Intestinal inflammation likely increased iron requirements in both WT and *Hamp* KO rats (to support the immune response and tissue repair), but this had differential influences on iron absorption depending upon the genotype (and resulting phenotype) of the animals. Thus, in contrast to WT rats, iron absorption decreased (in male KOs) or was unchanged (in female KOs) during acute colitis (Fig 22A and 22I). In males, distribution of freshly absorbed [59]Fe was largely uninfluenced by intestinal inflammation, except for a significant decrease in iron accumulation in the liver (Fig 22B–22H). In female KOs, [59]Fe distribution was similar in many tissues irrespective of intestinal inflammation (Fig 22C, 22D and 22F and 22G), but iron levels increased in blood (Fig 22B), bone (Fig 22E) and heart (Fig 22H). These data thus further support the concept that (patho)physiological responses to intestinal inflammation vary by sex and presence or absence of hepcidin (see Figs 23 and 24 for direct genotype comparisons), and according to level of body iron stores.

## Discussion

A rat model of acute colitis was utilized here to assess the impact of intestine-restricted inflammation on iron absorption. We hypothesized that localized intestinal

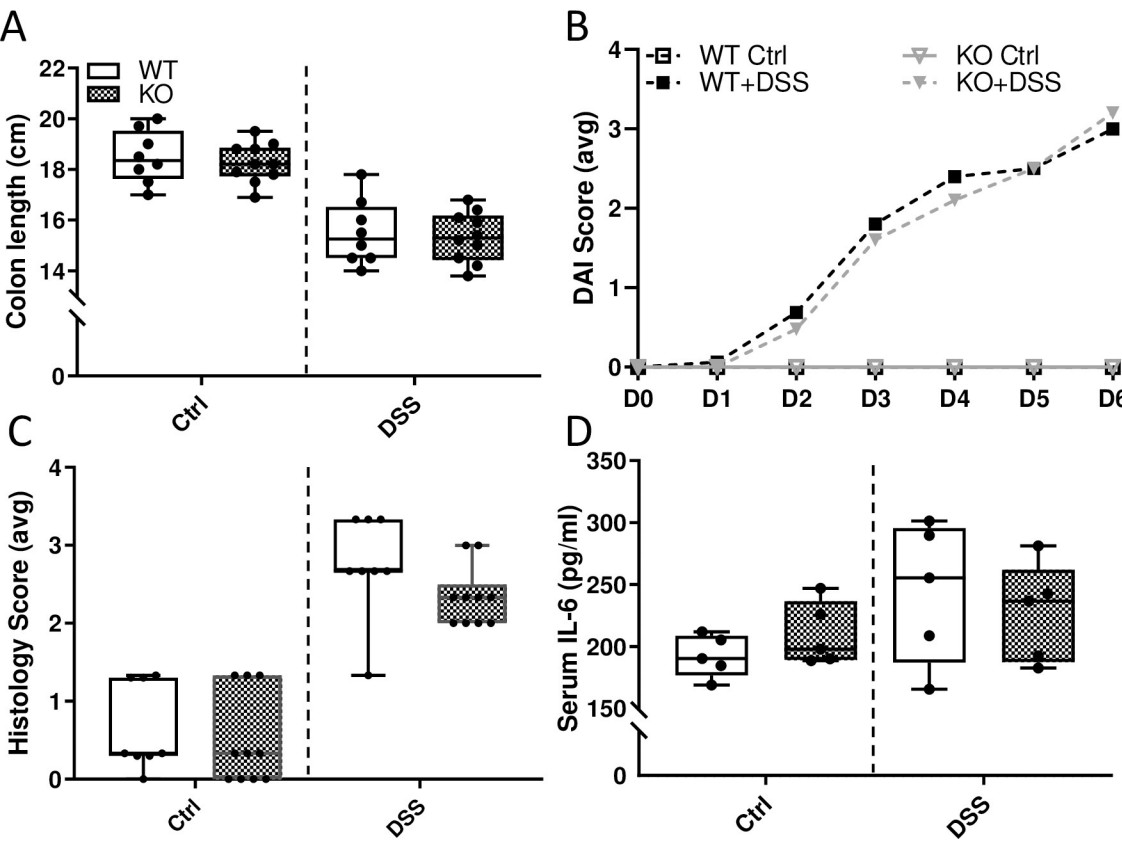

**Fig 18. Severity of DSS-induced colitis is similar between male WT and *Hamp* KO rats.** Length of the colon, defined as the end of the cecum to the anus, was measured as a macroscopic indicator of DSS-related colonic damage in 10-week-old rats (**A**). Data are presented as box pots for n = 7–11 rats/group and were analyzed separately by genotype using a t-test (no significant differences were noted). Disease activity index (DAI) scores were also determined daily (**B**). Scores were averaged on each day for each group. Mean values are shown for n = 7–11 rats/group. Data were analyzed by repeated measures 2-way ANOVA; no statistical differences were noted between genotypes. KO Ctrl versus KO + DSS, *p*<0.0001; WT Ctrl versus WT + DSS, *p*<0.0001. Histology scores from H&E-stained sections from descending colon and duodenum/proximal jejunum are shown in panel **C**. Sections from control and DSS-treated WT and *Hamp*[-/-] rats were blindly scored in 4 randomly selected areas. Each area was assigned a score from 0–4 based on levels of inflammatory cell infiltration, goblet cell depletion, and damage to the crypt/villus architecture. Data are presented a box plots for n = 4 rats/group and were analyzed separately by genotype using a t-test (no significant differences were noted). Further, serum IL-6 levels were assessed by ELISA (**D**). No significant differences were noted between control or DSS treatment groups or between genotypes (by t test), indicating a lack of systemic inflammation.

inflammation would increase overall iron demand, and that hepatic *Hamp* expression would be suppressed in response, thus upregulating iron absorption (and enhancing iron release from stores). The predicted regulatory loop invokes the 'stores' regulator of iron absorption [1, 2, 5], in which increased iron demand begins to deplete iron stores and cause hypoferremia, thus triggering a down regulation of hepatic *Hamp* expression. The mechanism probably relates to a HFE/TFR1/TFR2, BMP-SMAD signaling pathway in hepatocytes in which the level of serum iron is 'communicated' to the *Hamp* gene [30]. Also, given that acute colitis caused anemia in male WT rats, the 'erythroid' regulator of iron absorption [3] would theoretically also be involved in suppressing *Hamp* expression. Recent evidence suggests that developing erythrocytes produce and secrete a hormone, erythroferrone, which downregulates hepatic *Hamp* expression when erythroid demand for iron is elevated (thus increasing serum iron and potentiating iron delivery to the bone marrow) [33].

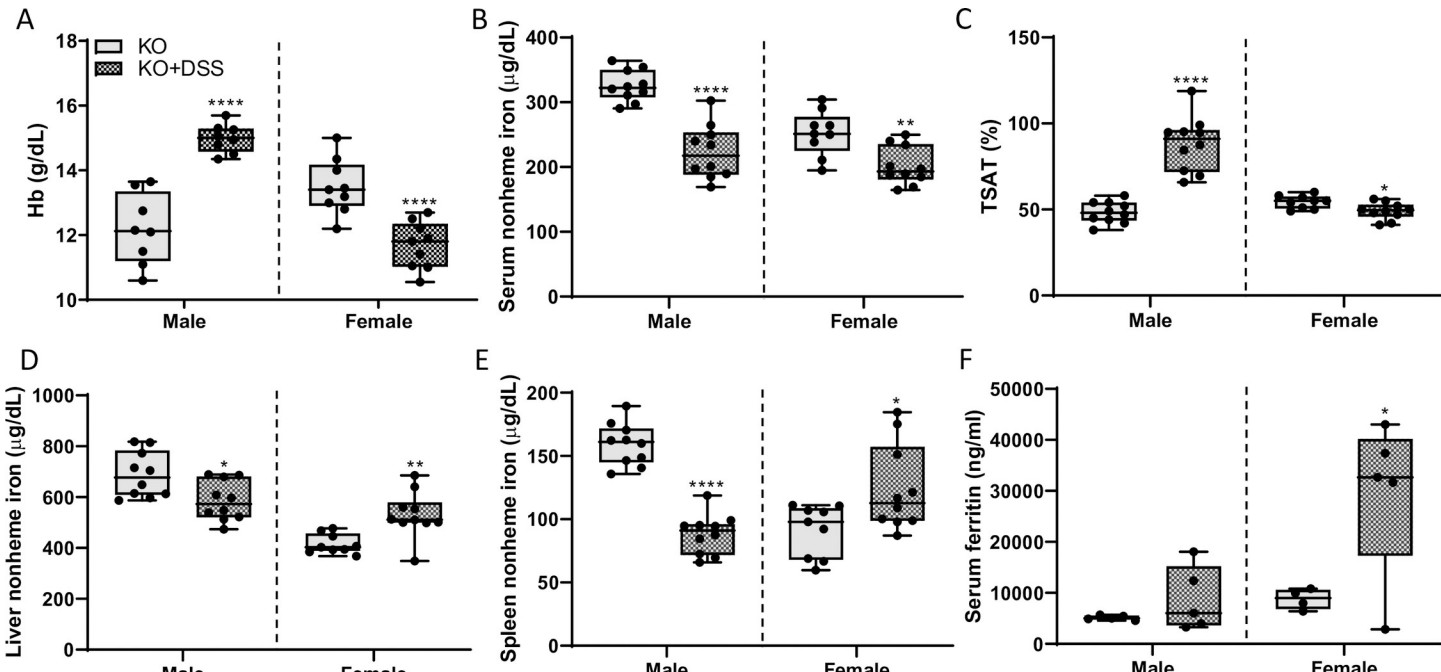

**Fig 19. Intestinal inflammation differentially influences iron homeostasis in male and female *Hamp*⁻/⁻ rats.** Hb levels were assessed in 10-week-old control and DSS-treated hepcidin KO rats (**A**). Serum nonheme iron (**B**), TSAT (**C**), liver (**D**) and spleen (**E**) nonheme iron concentrations, and serum ferritin levels (**F**) were also determined using standard methodologies. Data are presented as box plots for n = 5–11 rats/group and were analyzed separately by sex using a t test (*$p$ = 0.0349; **$p < 0.0034$; ***$p < 0.0006$; ****$p < 0.0001$).

Notably, in the current investigation, acute DSS treatment increased iron demand and thus influenced systemic iron homeostasis. For example, TSAT decreased, and iron was mobilized from the spleen in WT rats of both sexes, probably reflecting an increase in demand for iron to support enhanced erythropoiesis. Anemia and hypoferremia were only evident in males, consistent with more severe GI disease noted in males. Despite iron depletion and anemia, hepatic *Hamp* expression was unaffected, contrary to our predictions (and to current state of knowledge in the field, as outlined above). Nevertheless, iron absorption increased during acute colitis in both sexes of WT rats. Collectively, these findings demonstrated that iron absorption was appropriately upregulated during colitis (when iron demand increased), and further that this increase did not relate to suppression of hepatic *Hamp* expression. Therefore, to better understand regulation of iron absorption during acute intestinal inflammation, parallel studies were pursued in global *Hamp* KO rats. We postulated that: 1) the severity of colitis would not be exacerbated in the KOs, since (paradoxically) the intestinal epithelium is iron depleted due to high FPN1 iron export activity [32, 34–36]; 2) iron demand would be similarly increased in *Hamp*⁻/⁻ rats during acute intestinal inflammation (to support the immune response and tissue regeneration); and 3) increased demand for iron could be met (at least in part), by release of storage iron (which is at maximal capacity in these rats).

As part of this investigation, we established and characterized a new model of human hereditary hemochromatosis (HH), the *Hamp* KO SD rat. The central role of hepcidin in regulating iron absorption is perhaps best exemplified in this genetic disease [37]. Some forms of HH result from mutations in genes that regulate *Hamp* transcription in hepatocytes, including homeostatic iron regulator (HFE) [type 1 HH], hemojuvelin (HJV) [type 2A HH], and transferrin receptor 2 (TFR2) [type 3 HH]. In these types of HH, hepcidin expression is inappropriately low in relation to body iron burden. Mutations in *Hamp* underlie another, particularly

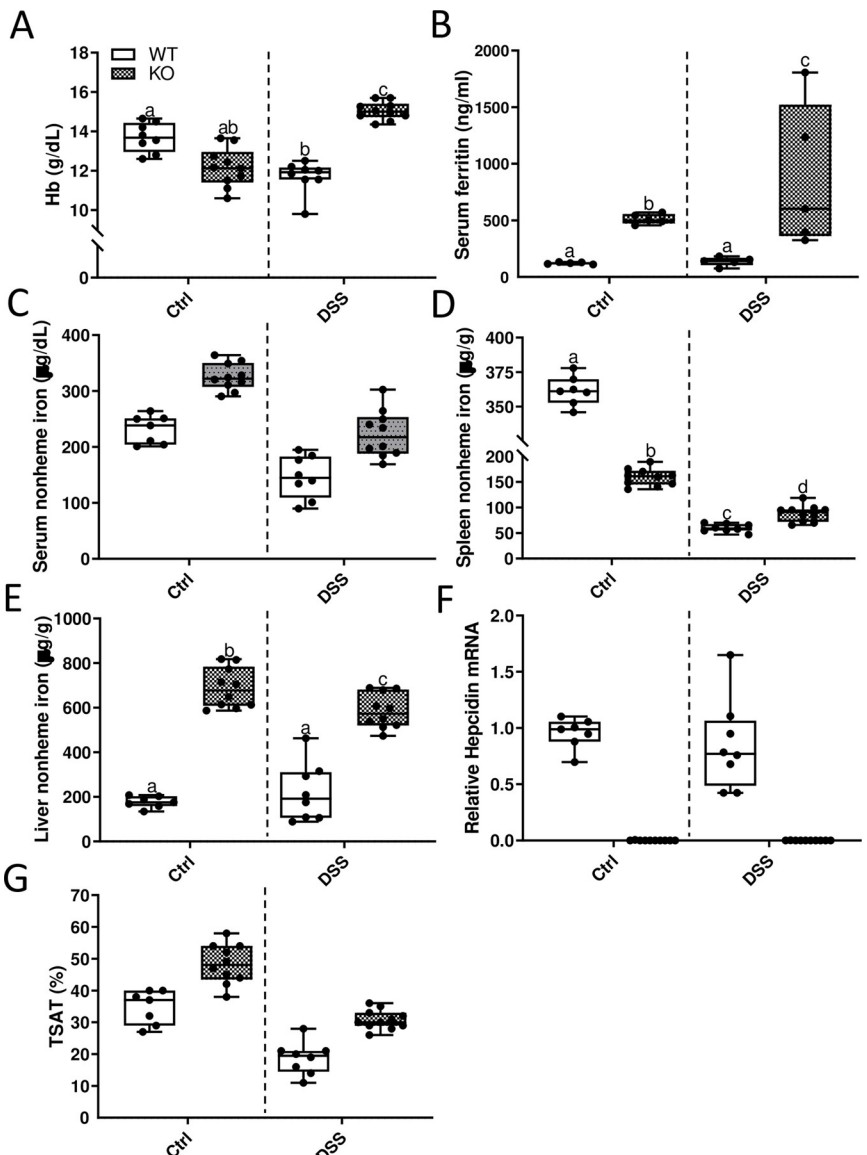

**Fig 20. DSS treatment has differential effects on blood Hb levels and spleen and liver iron content in male WT versus *Hamp* KO rats.** Hb levels were assessed in 10-week-old control and DSS-treated male WT and *Hamp*[-/-] rats (**A**). Serum ferritin levels were also determined (**B**). Serum (**C**), spleen (**D**) and liver (**E**) nonheme iron content was also assessed. Liver hepcidin mRNA expression was determined by qRT-PCR (**F**). TSAT was also assessed post-mortem (**G**). Data are presented as box plots for two independent experiments with n = 8–10 rats/group and were analyzed by two-way ANOVA with Tukey's multiple comparisons post-hoc test. Groups without common letters differ significantly ($p \leq 0.0101$). Genotype × Treatment, $p \leq 0.0001$ (**A, B, D, E**); Genotype $p \leq 0.0014$ (**A-E, G**); Treatment, $p \leq 0.0001$ (**C, D, G**).

severe form of the disease (type 2B HH) [38]. A detailed description of the different types of HH has been recently published [39]. When hepcidin expression is low (or absent), intestinal iron absorption is chronically elevated, leading to excessive body iron accumulation (and associated oxidative damage), possibly resulting in the development of liver fibrosis/cirrhosis, diabetes, arthropathies and endocrinopathies.

Most experimental models of HH are mice. The scientific rationale for using mice for this purpose is, however, unclear, especially since mice do not recapitulate some aspects of the

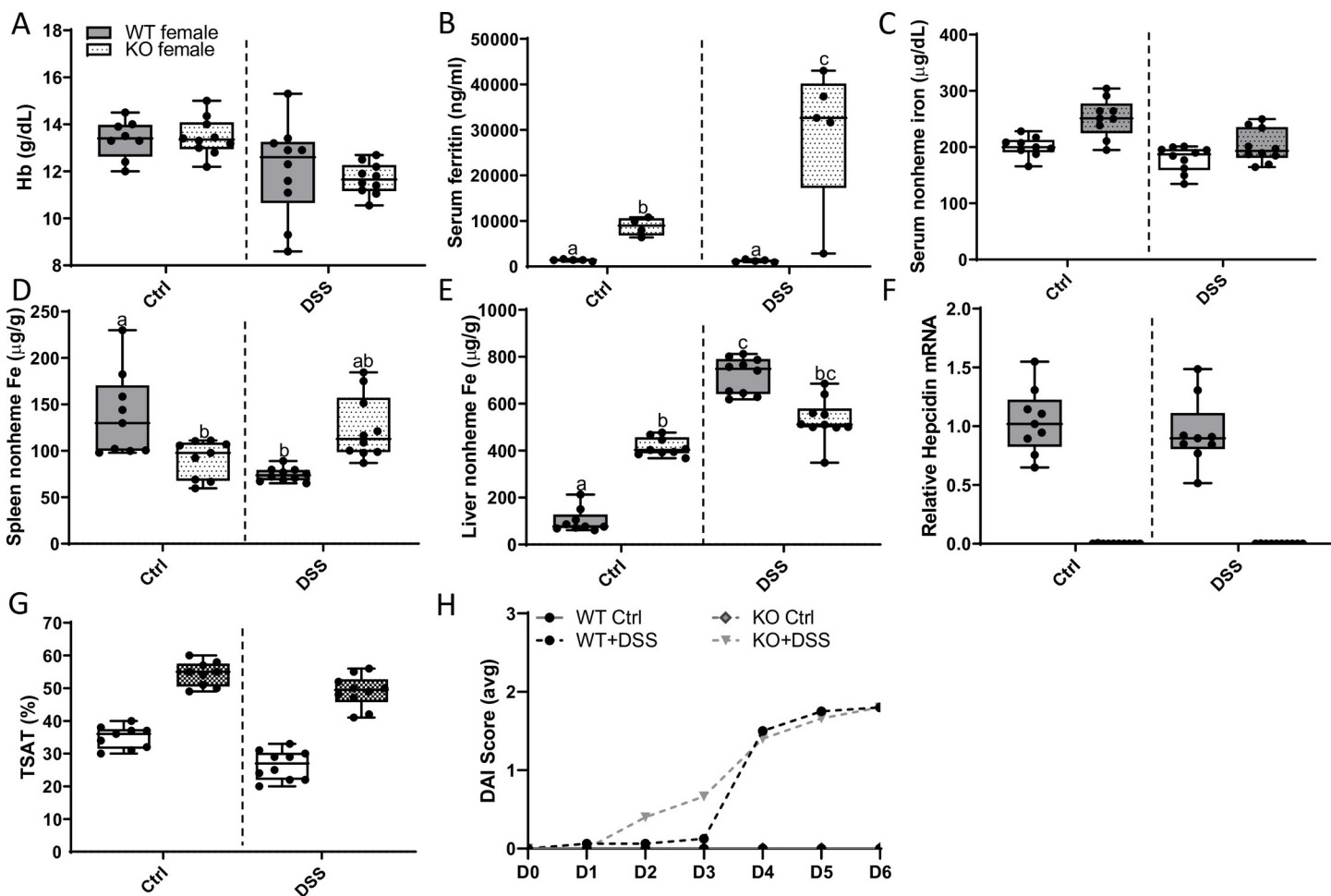

**Fig 21. DSS treatment had differential effects on blood Hb levels and spleen and liver iron content in female WT versus *Hamp* KO rats.** Blood hemoglobin (**A**) and serum ferritin (**B**) levels were determined at sacrifice. Serum (**C**), spleen (**D**), and liver (**E**) nonheme iron levels were assessed using standard methods. Liver *Hamp* expression was also quantified by qRT-PCR (**F**). TSAT was also assessed using routine methods (**G**). Data are presented as box plots for n = 8–10 rats/group and were analyzed by 2-way ANOVA followed by Tukey's multiple comparisons post hoc test. Groups labeled with different letters vary significantly ($p<0.05$). Genotype × Treatment, $p\leq0.0134$ (**B, D, E**); Genotype, $p = 0.0080$ (**B, D**); Treatment, $p\leq0.0147$ (**A-C, E, G**). Disease activity index (DAI) scores were determined daily in water (control) and DSS-treated rats of both genotypes (**H**). Scores were averaged on each day for each group. Data were analyzed by repeated-measures 2-way ANOVA (Treatment main effect, $p\leq0.0004$; KO Ctrl vs. KO + DSS, $p<0.0001$; WT Ctrl vs. WT + DSS, $p<0.0001$). No statistical differences were noted between genotypes.

human disease [40]. Also, importantly, humans consume heme and nonheme iron, and (inappropriately) increased absorption of both forms of dietary likely contributes to iron loading in HH [41]. Unfortunately, mice cannot utilize dietary heme iron [42], so this is a major limitation of existing mouse models of HH. We hypothesized that regulation of iron metabolism in rats could be more similar to what occurs in humans [43]. Moreover, we (and others) [44] have demonstrated that Sprague-Dawley rats can efficiently absorb and utilize heme iron (Flores, S.R.L, Collins, J.F., unpublished data). Our (yet unpublished) studies showed that heme iron can correct iron deficiency anemia in SD rats fed a low iron diet, and that a diet containing mainly (∼85%) heme as the iron source can support normal reproduction and growth and development of fetal and suckling neonatal rats. This then is a clear advantage (over mice) when modeling human iron-related disorders. This information collectively then summarizes our rationale for creating a SD rat model of HH. Another advantage of the Sprague-Dawley (SD) strain is that it is outbred, perhaps better reflecting human genetic diversity (than inbred

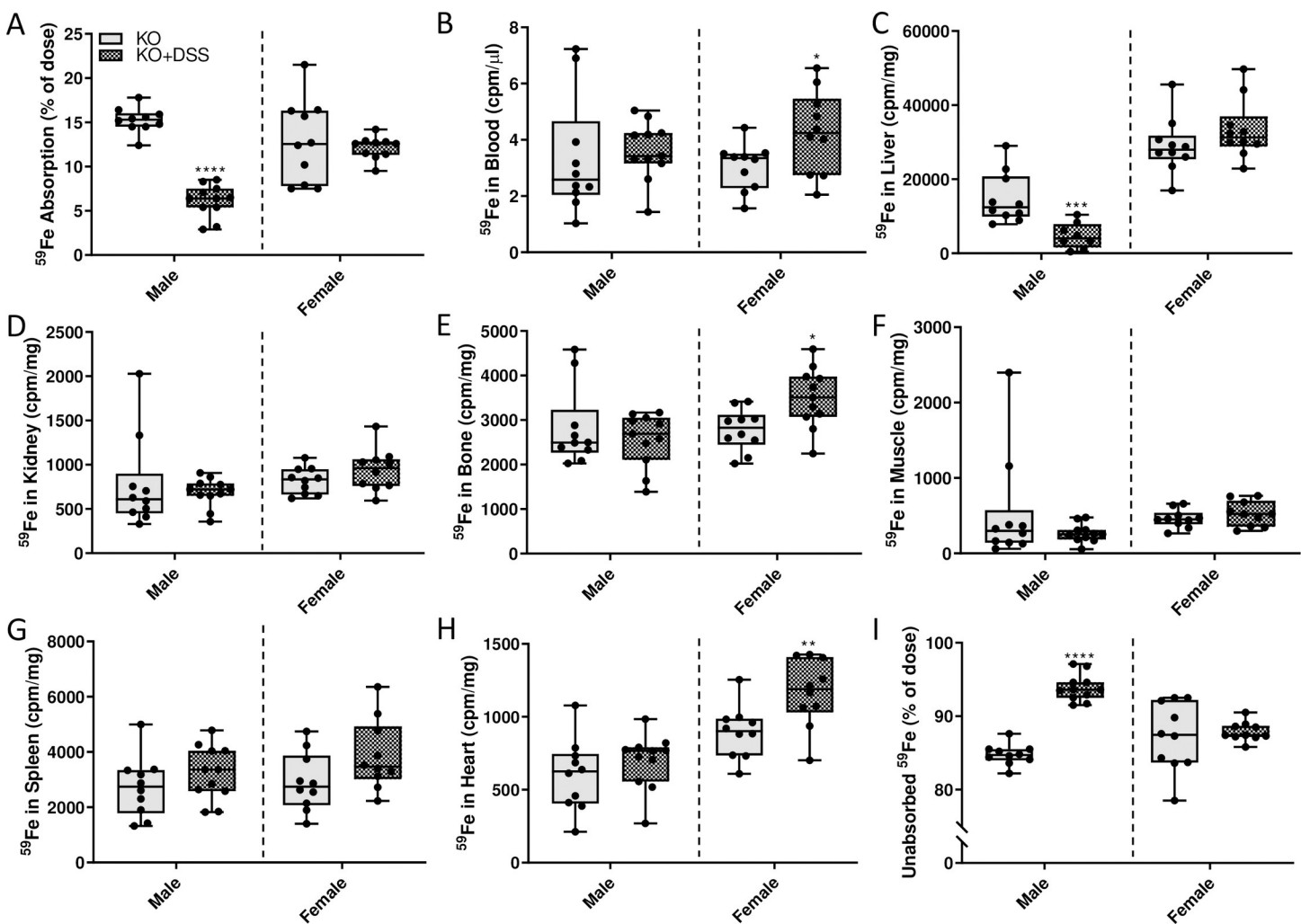

**Fig 22. Radiotracer iron absorption studies in male and female *Hamp*$^{-/-}$ rats.** Intestinal iron absorption and $^{59}$Fe tissue distribution was determined in control or DSS-treated, 10-week-old hepcidin KO rats of both sexes using a standard oral gavage protocol. Iron absorption, as percent of $^{59}$Fe radioactivity remaining in the animal 24 hours after dosing (minus radioactivity retained in the GI tract), is depicted in panel **A**. Distribution of $^{59}$Fe in several tissues was also assessed by gamma counting (**B**-**H**). The percentage of unabsorbed $^{59}$Fe is also depicted (**I**). All data are presented as box plots for 7–11 animals from two independent experiments and were analyzed separately by sex using a t-test (*$p<0.0331$; **$p<0.0092$; ***$p = 0.0001$; ****$p<0.0001$).

mouse strains). We chose to model type 2B HH (due to mutations in *Hamp*), which is a very rare, early onset form of the human disease [38], since we postulated that observations made in this more severe HH model would also be informative in relation to other (less severe) forms of the disease.

Early-stage pathological manifestations of type 2B HH include increases in serum and tissue nonheme iron concentrations; therefore, we assessed the temporal pattern of iron loading in *Hamp* KO rats from weaning to one year of age. *Hamp* KO rats had elevated serum nonheme iron and TSAT by 3 (in males) or 9 (in females) weeks of age, with progressive increases in these parameters noted up to one year of age. Many end-stage pathological manifestations of human HH (i.e., cirrhosis, diabetes, cardiomyopathy) occur only after significant, decades-long-lasting tissue iron accumulation. The tissue and organ dysfunction that typify human HH have been problematic to model in laboratory mice, perhaps since mice have a much shorter life span, or due to basic physiological differences between mice

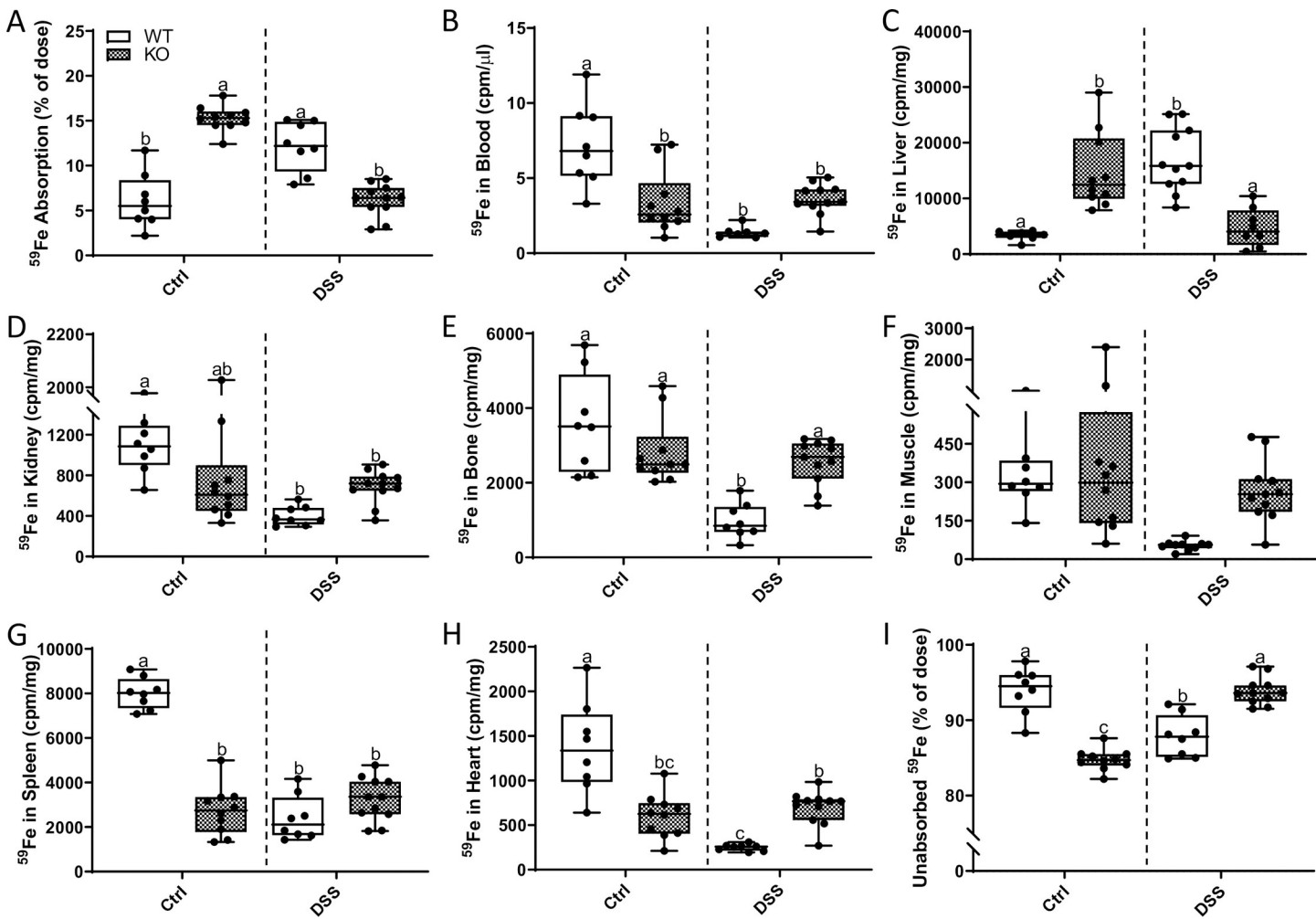

**Fig 23. DSS treatment differentially influences intestinal $^{59}$Fe absorption and tissue distribution in male WT and *Hamp* KO rats.** Intestinal iron absorption and tissue distribution were assessed in 10-week-old male rats of both genotypes with and without intestinal inflammation. The absorption of a test dose of $^{59}$Fe administered by oral, intragastric gavage is presented as the percentage of radioactivity present in the animals (minus $^{59}$Fe trapped in the gut) 24 h after dosing (**A**). Distribution of $^{59}$Fe in several tissues was also assessed by gamma counting (**B-H**). The percentage of unabsorbed $^{59}$Fe is also depicted (**I**). All data are presented as box plots for 7–10 animals from two independent experiments and were analyzed by 2-way ANOVA followed by Tukey's multiple comparisons post-hoc test. Groups labeled with different letters are significantly different from one another. Data that did not fit a normal distribution were transformed prior to performing statistical analyses. Genotype × Treatment, $p < 0.0063$ (**A-E, G-I**); Genotype $p \leq 0.0428$ (**A, G**); Treatment $p \leq 0.0428$ (**B, D, E-H**).

and humans [45]. In any case, it was important to track iron loading as the *Hamp*$^{-/-}$ rats aged. Excessive hepatic nonheme iron accumulation was observed by 4.5 weeks of age in females, and by 6 weeks of age in males, with progressive increases noted (relative to WT controls) out to one year of age in both sexes. Consistent with this, both sexes of adult KO rats had elevated serum ferritin levels. Early stages of liver fibrosis were documented in some older KOs. In *Hamp*$^{-/-}$ rats, liver iron was deposited with a periportal distribution reflecting parenchymal iron overload, a pattern that is also observed in humans with the most common type of HH (HFE-related HH) [46]. Interestingly, this pattern of iron loading is distinct from the centrilobular distribution first reported in *Hamp* KO mice [47]. Further- more, we previously documented a similar, (early life) chronology of iron loading in *Hamp* KO mice [48]. In sum, we have successfully established the *Hamp* KO rat as a new model of early on-set (or juvenile) HH.

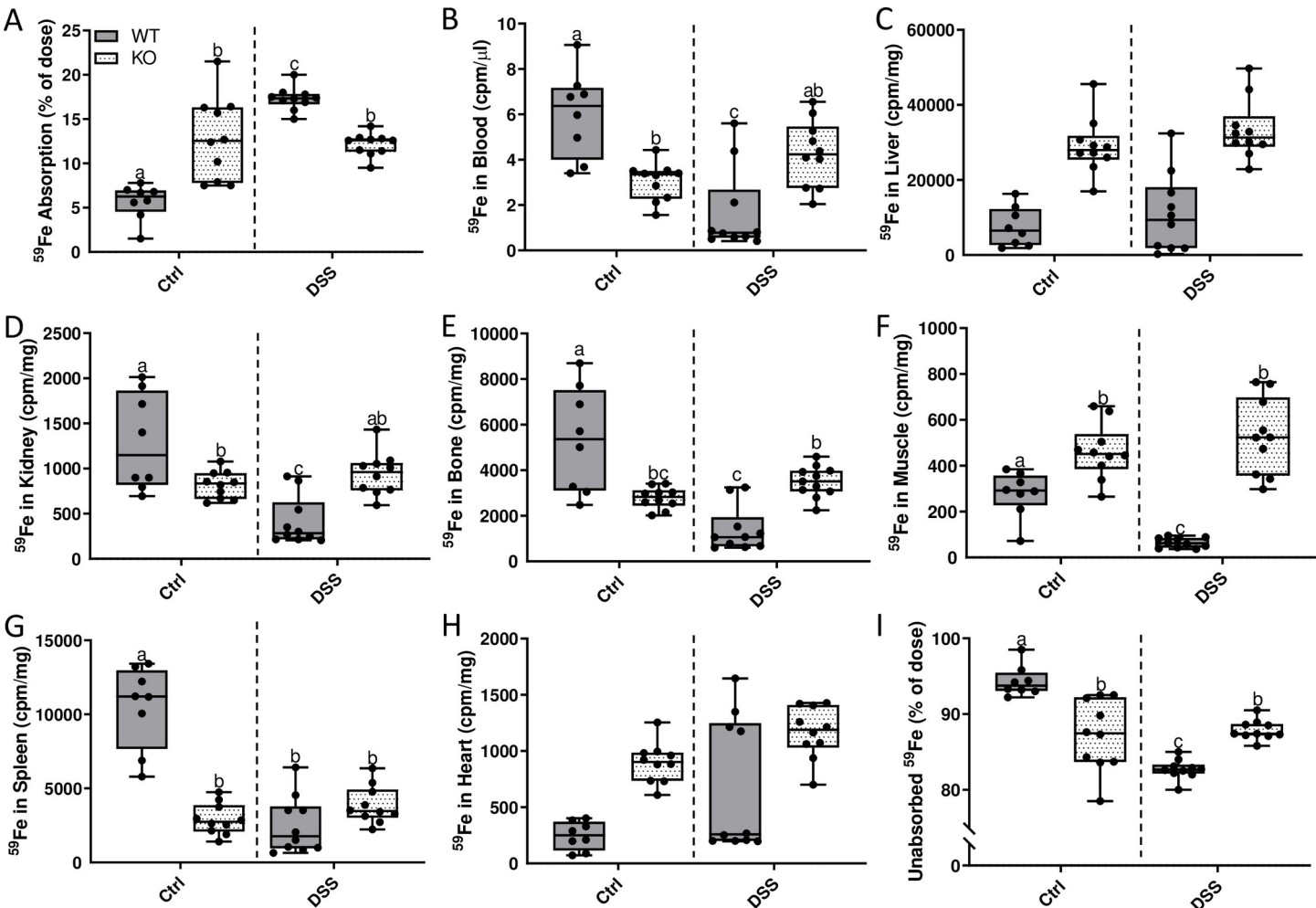

**Fig 24. DSS treatment differentially influences intestinal $^{59}$Fe absorption and tissue distribution in female WT and *Hamp* KO rats.** Intestinal iron absorption and $^{59}$Fe tissue distribution was determined using the transport assay described in the Materials and methods section (**A**). Distribution of $^{59}$Fe in several tissues was also assessed by gamma counting (**B-H**). The percentage of unabsorbed $^{59}$Fe is also depicted (**I**). All data are presented as box plots for 8–10 animals from two independent experiments and were analyzed by 2-way ANOVA followed by Tukey's multiple comparisons post-hoc test. Groups labeled with different letters are significantly different from one another ($p \leq 0.0328$). Genotype × Treatment $p < 0.0007$ (**A-B, D-G, I**), Genotype $p < 0.0001$ (**C, F, G-H**), Treatment $p \leq 0.0041$ (**B, D, E, H**).

In human HH, as transferrin saturation increases, non-transferrin bound iron (NTBI) appears in the blood and is taken up by various tissues and organs. Ultimately, this excess iron causes oxidative damage, leading to the characteristic pathologies associated with HH. In *Hamp*$^{-/-}$ rats, (possibly pathologic) iron accumulation was first observed in heart, pancreas, and kidney at 5–9 months of age (with sex differences noted). It seems logical that these tissues accumulate excess iron later, only after hepatic iron stores approach capacity and TSAT (and presumably NTBI) increases. Additionally, iron deposits were observed in the exocrine (i.e., in acini) and endocrine (i.e., in islets) pancreas in 9-month and one-year-old KOs. Iron accumulation in islets is consistent with human HH, which is typified by the development of diabetes (due to oxidative damage to insulin-producing β cells). While we did not detect abnormalities in glucose homeostasis in a few 9-12-month-old KO rats (unpublished observation), further studies are necessary to determine whether these rats develop overt diabetes later in life (note that rats can live up to 3.0+ years). Importantly, mice modeling human HH (e.g., *Hamp* KO mice) [49] load iron mainly in the exocrine pancreas, which has been a major limitation in this

area of research. Additionally, like what has been described in mouse models of HH (and in humans) [50], spleen iron was depleted in *Hamp* KO rats, as first noted in both sexes at 9 weeks of age. These observations are consistent with unregulated iron flux from splenic macrophages due to inappropriately high FPN1 activity. The age-related increase in splenic nonheme iron content seen in WT rats did not occur in KOs out to one year of age. Moreover, spleens from many older KO males were a lighter brown color (as compared to the normal deep red color) with a metallic sheen. These abnormal brown spleens also had an increased ratio of red to white pulp, possibly reflecting enhanced extramedullary erythropoiesis, which can occur in rats during anemia or iron-restricted erythropoiesis. Although older male KOs were not anemic, increased serum NTBI associated with elevated TSAT levels may cause RBC hemolysis, thus potentially increasing erythropoietic demand. Further functional and molecular analyses of these abnormal spleens are clearly warranted.

As anticipated, parenchymal tissue iron loading did not influence the severity of DSS-induced intestinal pathology in *Hamp* KO rats. This is consistent with what was observed during the acute phase of DSS exposure in a recent DSS study in *Hamp* KO mice [51]. Iron homeostasis in *Hamp* KOs was also influenced by acute colitis, but responses varied from WT rats. Sex differences were also noted for some parameters. For example, DSS exposure increased Hb levels in male KOs, but caused a minor reduction in females. Also, splenic and hepatic nonheme iron content decreased in KO males, but was not different in females. Moreover, colitis increased serum ferritin in female KOs (but not in males), most likely reflecting increases in body iron stores (since systemic inflammation, which increases serum ferritin, was not observed). These differential effects of DSS exposure could reflect the influence of the sex hormones on iron homeostasis, although disease severity was also less in females, which could explain some of the noted differential responses to DSS treatment. Further experimentation is required to elucidate sex-specific aspects of how acute DSS colitis, with localized intestinal inflammation, influences iron homeostasis.

Striking differences were noted when comparing how DSS exposure affected intestinal iron absorption in WT and *Hamp* KO rats. Iron absorption and liver $^{59}$Fe accumulation decreased (in male) or did not change (in female) *Hamp*$^{-/-}$ rats, as compared to noted increases in both sexes of DSS-treated WT rats. Consistent with unaltered or decreased $^{59}$Fe absorption, tissue distribution of $^{59}$Fe was largely unaffected during acute colitis in the KOs, again in sharp contrast to the situation in DSS-exposed WT rats. The logical conclusion from this series of experiments is that lack of hepcidin and/or tissue iron loading alters the physiological response to intestinal inflammation. The most likely possibility relates to the high liver iron levels in the KOs, which allows the increased physiological demand for iron to be met by release of hepatic storage iron (thus obviating a need to increase intestinal iron absorption, as in WT rats). Furthermore, the notable iron-related, pathophysiological disturbances caused by acute DSS exposure were probably not mediated by circulating hepcidin, as no change in hepatic *Hamp* expression was seen in WT rats (when iron absorption increased) and the KOs (obviously) do not produce any hepcidin (yet iron absorption decreased).

In conclusion, we have demonstrated that acute colitis in rats elevated iron demand but that this did not suppress *Hamp* expression (as would have been predicted). Nonetheless, intestinal iron transport increased during acute colitis with GI tract-restricted inflammation. Therefore, to better understand regulation of iron absorption during acute colitis, and the role that hepcidin may play in this process, parallel studies were performed in *Hamp* KO rats. Importantly, these experiments revealed that hepatic iron stores were mobilized, and iron absorption was concurrently suppressed in the KOs. This was somewhat surprising as: 1) hepcidin is thought to be the main regulator of iron absorption and iron release from stores; and 2) hepcidin cannot simultaneously increase iron release from stores and suppress intestinal

iron absorption since its sole mechanism of action relates to modulating the iron export activity of FPN1 (which is expressed on the basolateral membrane of duodenal enterocytes and also on the plasma membrane of cells that store iron in the liver [i.e., Kupffer cells and hepatocytes]). In summary then, this investigation has demonstrated that intestinal iron transport can be modulated during acute intestinal inflammation independent of changes in liver *Hamp* expression and when *Hamp* is ablated.

## Supporting information

**S1 File.**
(PDF)

**S1 Graphical abstract.**
(TIF)

## Author Contributions

**Conceptualization:** Didier Merlin, James F. Collins.

**Data curation:** Shireen R. L. Flores, Savannah Nelson, Regina R. Woloshun, Xiaoyu Wang, Jung-Heun Ha, Jennifer K. Lee, Yang Yu.

**Formal analysis:** Shireen R. L. Flores, Savannah Nelson, Regina R. Woloshun, Xiaoyu Wang, Jung-Heun Ha, Jennifer K. Lee, Yang Yu, Didier Merlin, James F. Collins.

**Investigation:** Regina R. Woloshun, Xiaoyu Wang, Jung-Heun Ha, Jennifer K. Lee, Yang Yu, James F. Collins.

**Methodology:** Shireen R. L. Flores.

**Project administration:** James F. Collins.

**Supervision:** James F. Collins.

**Validation:** Shireen R. L. Flores.

**Visualization:** Didier Merlin.

**Writing – original draft:** Shireen R. L. Flores.

**Writing – review & editing:** Shireen R. L. Flores, Savannah Nelson, Regina R. Woloshun, Xiaoyu Wang, Jung-Heun Ha, Jennifer K. Lee, Yang Yu, Didier Merlin, James F. Collins.

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
