## [Decision Letter · Decision Letter 0]

21 Jan 2021

PONE-D-20-38471

Intestinal iron absorption is appropriately modulated to match physiological demand for iron in wild-type and iron-loaded Hamp (hepcidin) knockout rats during acute colitis

PLOS ONE

Dear Dr. Collins,

Thank you for submitting your manuscript to PLOS ONE. After careful consideration, we feel that it has merit but does not fully meet PLOS ONE’s publication criteria as it currently stands. Therefore, we invite you to submit a revised version of the manuscript that addresses the points raised during the review process.

The revised manuscript should address the issues raised by both reviewers regarding structure, organization, inconsistencies and errors. The rationale for generating the rat hepcidin knockout model and the advantages offered by it should be outlined. The data should also be discussed in the context of relevant recent literature.

We look forward to receiving your revised manuscript.

Kind regards,

Kostas Pantopoulos, PhD

Academic Editor

PLOS ONE

Journal Requirements:

2. As part of your revisions, please revise your Methods section to address the following:

(i) the number of animals in each group and how you determined the sample size;

(ii) the sex and strain of the mice;

(iii) if applicable: all anesthetics and analgesics administered to animals during your study (name of drug, dosage, frequency and route of administration);

(iv) details about humane endpoints for any animals who became severely ill during the study;

(v) the rate of mortality during the study and the cause of death (if applicable);

Lastly, please complete and submit the ARRIVE Guidelines checklist (Essential 10 version): https://arriveguidelines.org/resources/author-checklists

4. Please ensure that you refer to Figure 7 in your text as, if accepted, production will need this reference to link the reader to the figure.

Reviewers' comments:

Reviewer's Responses to Questions

**Comments to the Author**

1. Is the manuscript technically sound, and do the data support the conclusions?

Reviewer #1: Partly

Reviewer #2: Yes

2. Has the statistical analysis been performed appropriately and rigorously? 

Reviewer #1: Yes

Reviewer #2: Yes

3. Have the authors made all data underlying the findings in their manuscript fully available?

Reviewer #1: Yes

Reviewer #2: Yes

4. Is the manuscript presented in an intelligible fashion and written in standard English?

Reviewer #1: Yes

Reviewer #2: No

5. Review Comments to the Author

Reviewer #1: In this manuscript, Flores et al. showed that “intestinal iron absorption is appropriately modulated to match physiological demand for iron in wild-type and iron-loaded Hamp (hepcidin) knockout rats during acute colitis”.

This manuscript describes for the first time the generation of hepcidin KO rats. Moreover, while hepcidin has been discovered as the key regulator of iron absorption, the authors show that intestinal iron transport can be modulated during acute intestinal inflammation independently of changes in liver hepcidin expression or deletion of hepcidin. Overall the manuscript is supported by many experiments with appropriate controls and statistical analyses. Figures are mostly clear and experimental methods well described.

- If this is the first description of a hepcidin KO in rats (?), it should be put forward as well as the advantages of this model. What is the rationale behind the choice of using rats rather than mice, given that Hepcidin KO mice are available ?

- The authors showed that, during acute colitis, hepatic iron stores were mobilized and iron absorption concurrently suppressed in the hepcidin KO mice and conclude that “regulatory mechanisms may not involve hepcidin”. However, in a model of hepcidin deficiency (inducible hepc KO model, use of hepcidin inhibitors, ..) with no elevated hepatic iron storage (closer to the clinical reality) , hepatic iron stores would not have been mobilized and the impact on iron absorption may have been different. Could the authors comment on that ?

- The manuscript could be reorganized to increase the clarity. It is useful to have data from WT and KO on the same graphs and it could be incorporated in the main manuscript .

A scheme or a table recapitulating the differences between males and females could be useful.

- When is radioactive iron given to mice (day 5 ? ).

- Existing literature is not properly taken into account, especially the recent article by Bessman et al., Science (2020), showing, that, similarly to the findings of the authors : In conditions of severe DSS exposure , WT and Hepcidin KO mice present similar inflammation in the acute phase (day 1 to Day 6) . This result should be presented in the manuscript and discussed properly.

Minor points :

- Fig.2A-B. Same magnification should be used

- Typo line 80, page 4 : Chron

- Paragraph “intestinal iron absorption decreases in male HampKO rats but is unchanged in females. “ is related to figure 7 and not figure 6 : lines 419-421 pages 19.

Reviewer #2: The authors sought to investigate the impact of localized intestinal inflammation on iron homeostasis, using WT and Hamp KO rat models. They found that acute gut inflammation had distinct effects on iron homeostasis that depend on the sex, genotype, and phenotype of the experimental rats. Iron absorption was modulated independently of Hamp transcription by an unknown mechanism.

Major comments :

The authors need to thoughtfully review the manuscript regarding the figures and their legends (both figure legends and graph legends), and their numbering. Several figures are missing or mislabeled.

Minor comments:

1. Please add reference(s) for the statement in lines 87-90.

2. The statement in the abstract (lines 48-51; sex and genotype) seems to be inconsistent with the statement at the end of the Introduction section (lines 100-103).

3. Section M&M – please do not use the abbreviation “SD rats” in line 130, since you already use SD for Standard Deviation.

4. M&M – line 205 – please specify which tissues and organs.

5. Results, Line 219-220 and line 345 – should refer to all Figure S1, that is “Fig. S1-A-D”?

6. Results, Line 240, your t-test symbols are inconsistent. Please specify t tests used.

7. Results, legend Fig 1, Line 247, please check P value and stars for consistency with the other graphs in the figure – should be three stars for p<0.001?

8. Results, Legend to Figure 2, Line 252-257 - inconsistent labeling and legend – Is it A, B, C, D, E, H? C and D missing from the Figure?

9. Line 324-326, please justify the KO line mutation choice.

10. Line 330, figure S7, reference to male in figure’s legend is missing (present in graph legend)

11. Line 333-336, please check reference to figures S8 and S9 - should you include also S7?

12. Line 345, Figure 2C D are missing

13. Line 415, figure S18 is missing from the manuscript

14. Line 417 and 419-421, do you mean figure 7 (not figure 6)?

6. PLOS authors have the option to publish the peer review history of their article (what does this mean?). If published, this will include your full peer review and any attached files.

Reviewer #1: No

Reviewer #2: No

---

## [Author Response · Author response to Decision Letter 0]

6 Apr 2021

3/15/21

RE: Resubmission of PONE-D-20-38471

Dear Academic Editor and Reviewers:

We wish to take this opportunity to thank the reviewers and associate editor for their thorough and thoughtful review of our manuscript. We have further revised this paper in response to one reviewer’s additional comments. A specific, point-by-point response to the reviewer’s comments follows below. 

Response to Reviewer #1:

General Comments: This manuscript describes for the first time the generation of hepcidin KO rats. Moreover, while hepcidin has been discovered as the key regulator of iron absorption, the authors show that intestinal iron transport can be modulated during acute intestinal inflammation independently of changes in liver hepcidin expression or deletion of hepcidin. Overall the manuscript is supported by many experiments with appropriate controls and statistical analyses. Figures are mostly clear and experimental methods well described.

Comment 1-1: - If this is the first description of a hepcidin KO in rats (?), it should be put forward as well as the advantages of this model. What is the rationale behind the choice of using rats rather than mice, given that Hepcidin KO mice are available?

Author Response 1-1: As requested by the reviewer, we have “put forward” all information concerning development and characterization of our new rat model of HH, the Hamp KO rat. This includes: 1) adding new Methods sections, one on generation of the Hamp KO rat and one on histological analysis of liver and spleen; 2) adding new Results sections describing the successful generation of the Hamp KO rat and characterization of the iron-overload phenotype of this new model as well as many new figures/legends; and 3) including an expanded discussion which provides the rationale for generating the Hamp KO rat and compares/contrasts the iron-loading phenotype with murine and human HH. 

Comment 1-2: The authors showed that, during acute colitis, hepatic iron stores were mobilized and iron absorption concurrently suppressed in the hepcidin KO mice and conclude that “regulatory mechanisms may not involve hepcidin”. However, in a model of hepcidin deficiency (inducible hepc KO model, use of hepcidin inhibitors, ..) with no elevated hepatic iron storage (closer to the clinical reality) , hepatic iron stores would not have been mobilized and the impact on iron absorption may have been different. Could the authors comment on that?

Author Response 1-2: In the scenario brought up by the reviewer, whereby hepcidin expression is repressed and liver iron content is normal, yes, we would predict that outcomes on intestinal iron absorption would be different. The influence on iron absorption may depend upon the rate and extent of liver iron depletion (to meet increased demand for iron associated with colitis), and when (or if) this leads to a decrease in transport iron and eventually iron-restricted erythropoiesis (with possible concurrent hypoxia). Given this complex scenario, outcomes could be difficult to predict. Therefore, given the quite speculative nature of this discussion and since we are unsure of how to weave this info into the manuscript, no changes have been made to the text in response to this reviewer comment. 

Comment 1-3: The manuscript could be reorganized to increase the clarity. It is useful to have data from WT and KO on the same graphs and it could be incorporated in the main manuscript.

Author Response 1-3: In retrospect, we can see how the paper was a bit confusing given the large amount of supplemental material (that may or may not be thoughtfully considered by many readers [or reviewers]). So, based upon comments from both reviewers regarding some confusion with how the paper was organized, we have eliminated all supplemental materials and incorporated all data into the main text of the manuscript. This includes adding figures to the main text showing direct genotype comparisons for many of the experiments performed as part of this investigation. 

Comment 1-4: A scheme or a table recapitulating the differences between males and females could be useful.

Author Response 1-4: The manuscript is quite lengthy and contains over 20 figures and one table already, so adding such additional information may not be ideal. Moreover, given that all data are now included within the main text, including genotype and sex comparisons, it should now be easier for the reader to appreciate these genotype or sex differences. So, based on this argument (which we hope the reviewer will agree with), we have not made changes to the paper. 

Comment 1-5: When is radioactive iron given to mice (day 5? ).

Author Response 1-5: The rats were fasted overnight on day 5 of DSS treatment and then given 59Fe on day 6 of DSS treatment. This has been clarified in the Methods section on “Quantification of intestinal iron absorption in experimental rats”. 

Comment 1-6: Existing literature is not properly taken into account, especially the recent article by Bessman et al., Science (2020), showing, that, similarly to the findings of the authors: In conditions of severe DSS exposure , WT and Hepcidin KO mice present similar inflammation in the acute phase (day 1 to Day 6) . This result should be presented in the manuscript and discussed properly.

Author Response 1-6: Yes, we are (of course) aware of this paper. It was, however, difficult for us to see how this study directly relates to the current investigation, for the following reasons: First, this study was performed in inbred mice and here, we have used a novel outbred rat model; Second, the DSS-exposure regimen was quite different in this mouse study as compared to our rat study (e.g., 2.5 or 3.5% DSS in mouse study, 4% in ours; exposure was until one group lost 10% body weight in the mouse study, ours was a 7 day exposure time and weight loss was minimal; in the mouse study, animals were sacrificed sometime after the DSS was removed from the drinking water, while in our rat study, animals were killed while still being exposed to DSS.; Third, mice and rats could very well respond differently to DSS exposure. This becomes apparent when comparing the responses in the mouse study to our rat study. In mice, Hamp ablation increased disease severity, while in rats, we documented no differences in disease severity between genotypes; And finally, in our rat study, we assessed several iron-related biomarkers and also quantified intestinal iron absorption, while the mouse study mainly focused on the immune response and tissue repair. Therefore, given these (and other) major differences in experimental approach and design, and overall focus and intent, we find it quite difficult to make meaningful comparisons between the two studies. Any attempt to do so would be quite speculative. However. given that the reviewer did point out one similarity between these studies, we have briefly raised this issue in the discussion section and cited this paper, as requested. 

Comment 1-7: Fig.2A-B. Same magnification should be used

Author Response 1-7: We tried to use the same magnification, but due to the lower quality of many of our images from these experiments, we were unable to do so. It is unclear what caused this, but unfortunately, this is the case. 

Comment 1-8: Typo line 80, page 4: Chron

Author Comment 1-8: This has been corrected. 

Comment 1-9: Paragraph “intestinal iron absorption decreases in male Hamp KO rats but is unchanged in females. “is related to figure 7 and not figure 6: lines 419-421 pages 19.

Author Response 1-9: This has been corrected.

Response to Reviewer #2:

General Comments: The authors sought to investigate the impact of localized intestinal inflammation on iron homeostasis, using WT and Hamp KO rat models. They found that acute gut inflammation had distinct effects on iron homeostasis that depend on the sex, genotype, and phenotype of the experimental rats. Iron absorption was modulated independently of Hamp transcription by an unknown mechanism.

Comment 2-1: The authors need to thoughtfully review the manuscript regarding the figures and their legends (both figure legends and graph legends), and their numbering. Several figures are missing or mislabeled.

Author Response 2-1: We have carefully edited the manuscript and all these issues have been corrected. 

Comment 2-2: Please add reference(s) for the statement in lines 87-90.

Author Response 2-2: We have added two references to AI and IBD as requested, but we are unaware of any studies of iron absorption/homeostasis in models of localized intestinal inflammation in the absence of a systemic inflammatory response (that we can cite). 

Comment 2-3: The statement in the abstract (lines 48-51; sex and genotype) seems to be inconsistent with the statement at the end of the Introduction section (lines 100-103).

Author Response 2-3: The key parts of the sentences referred to here are: 

“… in both sexes and genotypes of rats, iron absorption was appropriately modulated to match physiological demand for dietary iron during acute intestinal inflammation….”

“…acute, localized intestinal inflammation had distinct effects on iron absorption (and iron homeostasis), depending upon the sex and genotype/phenotype of experimental rats…”

We fail to see the contradiction with these statements. It would have been helpful for the reviewer to expound upon his/her concerns here, so that we could adequately address the specific perceived issues. 

Comment 2-4: Section M&M – please do not use the abbreviation “SD rats” in line 130, since you already use SD for Standard Deviation.

Author Response 2-4: “SD” had been changed to “Sprague-Dawley” 

Comment 2-5: M&M – line 205 – please specify which tissues and organs.

Author Response 2-5: Done. 

Comment 2-6: Results, Line 219-220 and line 345 – should refer to all Figure S1, that is “Fig. S1-A-D”?

Author Response 2-6: All figures and call outs within the text have been carefully edited. This has been corrected. 

Comment 2-7: Results, Line 240, your t-test symbols are inconsistent. Please specify t tests used.

Author Response 2-7: These issues have been corrected/amended.

Comment 2-8: Results, legend Fig 1, Line 247, please check P value and stars for consistency with the other graphs in the figure – should be three stars for p<0.001?

Author Response 2-8: This issue has been corrected. 

Comment 2-9: Results, Legend to Figure 2, Line 252-257 - inconsistent labeling and legend – Is it A, B, C, D, E, H? C and D missing from the Figure?

Author Response 2-9: This issue has been corrected.

Comment 2-10: Line 324-326, please justify the KO line mutation choice.

Author Response 2-10: The text has been expended on this issue- see text just after new Fig, 2. 

Comment 2-11: Line 330, figure S7, reference to male in figure’s legend is missing (present in graph legend)

Author Response 2-11: This issue has been corrected.

Comment 2-12: Line 333-336, please check reference to figures S8 and S9 - should you include also S7?

Author Response 2-12: This issue has been corrected.

Comment 2-13: Line 345, Figure 2C D are missing

Author Response 2-139: This issue has been corrected.

Comment 2-14: Line 415, figure S18 is missing from the manuscript

Author Response 2-14: This issue has been corrected.

Comment 2-15: Line 417 and 419-421, do you mean figure 7 (not figure 6)?

Author Response 2-15: This issue has been corrected.

Additional information regarding blot images: the original images of the dot blots shown in Figure 14 are included as Supplemental Information. 

Thanks again for the insightful and thoughtful review of our paper. We are hopeful that our extensive revisions have allayed most of the reviewer’s and associate editor’s concerns and that this novel paper can be expeditiously accepted for publication in Plos One. 

Sincerely,

James F. Collins, Ph.D. (on behalf of all authors)

Professor

Food Science & Human Nutrition Department

University of Florida

---

## [Decision Letter · Decision Letter 1]

19 May 2021

PONE-D-20-38471R1

Intestinal iron absorption is appropriately modulated to match physiological demand for iron in wild-type and iron-loaded Hamp (hepcidin) knockout rats during acute colitis

PLOS ONE

Dear Dr. Collins,

Thank you for submitting your manuscript to PLOS ONE. Both reviewers agree that the revised manuscript is improved and addresses all critical issues. Before final acceptance, we invite you to correct typos identified by reviewer 2 and submit a revised version.

We look forward to receiving your revised manuscript.

Kind regards,

Kostas Pantopoulos, PhD

Academic Editor

PLOS ONE

Journal Requirements:

Reviewers' comments:

Reviewer's Responses to Questions

**Comments to the Author**

1. If the authors have adequately addressed your comments raised in a previous round of review and you feel that this manuscript is now acceptable for publication, you may indicate that here to bypass the “Comments to the Author” section, enter your conflict of interest statement in the “Confidential to Editor” section, and submit your "Accept" recommendation.

Reviewer #1: All comments have been addressed

Reviewer #2: (No Response)

2. Is the manuscript technically sound, and do the data support the conclusions?

Reviewer #1: Yes

Reviewer #2: Yes

3. Has the statistical analysis been performed appropriately and rigorously? 

Reviewer #1: Yes

Reviewer #2: Yes

4. Have the authors made all data underlying the findings in their manuscript fully available?

Reviewer #1: Yes

Reviewer #2: Yes

5. Is the manuscript presented in an intelligible fashion and written in standard English?

Reviewer #1: Yes

Reviewer #2: Yes

6. Review Comments to the Author

Reviewer #1: The authors have adequately answered my questions or argued the lack of changes.

Reviewer #2: The quality of the manuscript has significantly improved. However, some minor issues are still present (typo errors).

Figure 13 - labeling of the figure should be corrected to macth the text and respect the order (A-B-C-D).

7. PLOS authors have the option to publish the peer review history of their article (what does this mean?). If published, this will include your full peer review and any attached files.

Reviewer #1: No

Reviewer #2: No

---

## [Author Response · Author response to Decision Letter 1]

24 May 2021

5/24/21

RE: Second Resubmission of PONE-D-20-38471

Dear Academic Editor and Reviewers:

We wish to take this opportunity to thank the reviewers and associate editor for their thorough and thoughtful review of our manuscript. We have further revised this paper in response to one reviewer’s additional comments. A specific, point-by-point response to the reviewer’s comments follows below. 

Response to Reviewer #2:

Comment 2-1: The quality of the manuscript has significantly improved. However, some minor issues are still present (typo errors).

Author Response Comment 2-1: We have thoroughly edited the entire manuscript and have made every effort to correct all typos and inconsistencies throughout the text. 

Comment 2-2: Figure 13 - labeling of the figure should be corrected to macth the text and respect the order (A-B-C-D).

Author Response Comment 2-2: Thank you for pointing out this error- this has been corrected.

Thanks again for the insightful and thoughtful review of our paper. We are hopeful that our extensive revisions have allayed most of the reviewer’s and associate editor’s concerns and that this novel paper can be expeditiously accepted for publication in Plos One. 

Sincerely,

James F. Collins, Ph.D. (on behalf of all authors)

Professor

Food Science & Human Nutrition Department

University of Florida

---

## [Editor Report · Decision Letter 2]

27 May 2021

Intestinal iron absorption is appropriately modulated to match physiological demand for iron in wild-type and iron-loaded Hamp (hepcidin) knockout rats during acute colitis

PONE-D-20-38471R2

Dear Dr. Collins,

We’re pleased to inform you that your manuscript has been judged scientifically suitable for publication and will be formally accepted for publication once it meets all outstanding technical requirements.

Kind regards,

Kostas Pantopoulos, PhD

Academic Editor

PLOS ONE
---

## [Editor Report · Acceptance letter]

7 Jun 2021

PONE-D-20-38471R2 

Intestinal iron absorption is appropriately modulated to match physiological demand for iron in wild-type and iron-loaded *Hamp* (hepcidin) knockout rats during acute colitis 

Dear Dr. Collins:

I'm pleased to inform you that your manuscript has been deemed suitable for publication in PLOS ONE. Congratulations! Your manuscript is now with our production department. 

Kind regards, 

on behalf of

Dr. Kostas Pantopoulos 

Academic Editor

PLOS ONE